# Circadian programming of the ellipsoid body sleep homeostat in *Drosophila*

Tomas Andreani[1], Clark Rosensweig[1], Shiju Sisobhan[1], Emmanuel Ogunlana[1], William Kath[2], Ravi Allada[3]*

[1]Department of Neurobiology, Northwestern University, Evanston, United States; [2]Department of Engineering Sciences and Applied Mathematics, Northwestern University, Evanston, United States; [3]Department of Neurobiology, Northwestern University, Chicago, United States

**Abstract** Homeostatic and circadian processes collaborate to appropriately time and consolidate sleep and wake. To understand how these processes are integrated, we scheduled brief sleep deprivation at different times of day in *Drosophila* and find elevated morning rebound compared to evening. These effects depend on discrete morning and evening clock neurons, independent of their roles in circadian locomotor activity. In the R5 ellipsoid body sleep homeostat, we identified elevated morning expression of activity dependent and presynaptic gene expression as well as the presynaptic protein BRUCHPILOT consistent with regulation by clock circuits. These neurons also display elevated calcium levels in response to sleep loss in the morning, but not the evening consistent with the observed time-dependent sleep rebound. These studies reveal the circuit and molecular mechanisms by which discrete circadian clock neurons program a homeostatic sleep center.

*For correspondence:
r-allada@northwestern.edu

Competing interest: The authors declare that no competing interests exist.

## Editor's evaluation

This work provides mechanistic insight into the interaction between the circadian and homeostatic systems that regulate sleep. Prolonged wakefulness can increase the need to sleep regardless of the time of day, so the question is whether the circadian clock influences this homeostatic regulation. The authors show that this build-up of sleep need does indeed vary with time of day under the control of specific clock neurons.

## Introduction

The classic two process model posits that the circadian clock and the sleep homeostat independently regulate sleep (*Borbély, 1982*; *Borbély et al., 2016*). The circadian process, via phased activity changes in central pacemaker neurons, times and consolidates sleep-wake (*Patke et al., 2020*). The less well-understood homeostatic process, often assayed after extended sleep deprivation, promotes sleep length, depth, and amount as a function of the duration and intensity of prior waking experience (*Deboer and Tobler, 2000*; *Franken et al., 1991*; *Huber et al., 2004*; *Werth et al., 1996*). Sleep homeostasis is thought to be mediated by the accumulation of various wake-dependent factors, such as synaptic strength (*Tononi and Cirelli, 2014*), which are subsequently dissipated with sleep.

While homeostatic drive persists in the absence of a functioning circadian clock (*Tobler et al., 1983*), homeostatic drive can be modulated by the circadian clock. Abolishing clock output through mutation of most core clock genes (*Franken et al., 2006*; *Laposky et al., 2005*; *Wisor et al., 2002*) or electrolytic ablation of the mammalian circadian pacemaker, the suprachiasmatic nuclei (SCN) (*Easton et al., 2004*) reduces SD-induced changes in non-rapid eye movement (NREM) sleep, an indicator of homeostatic sleep drive in mammals. As circadian clock genes and even the SCN may regulate

processes that are not themselves rhythmic (*Fernandez et al., 2014*), these studies leave open the question about whether homeostasis is circadian regulated. To more definitely address the interaction between the clock and the homeostat, sleep-wake have been scheduled to different circadian times in forced desynchrony protocols (*Dijk and Czeisler, 1994*; *Dijk and Czeisler, 1995*). In one such protocol, sleep and wake are scheduled to occur every 28 hr, allowing the circadian clock to free-run with a~24 hr period. Under these conditions, a variety of indicators of homeostatic drive such as total time asleep, latency to sleep, and NREM sleep time are reduced in the evening independently of time awake (*Dijk and Czeisler, 1994*; *Dijk and Czeisler, 1995*; *Dijk and Duffy, 1999*; *Lazar et al., 2015*), consistent with the idea that the clock sustains wakefulness at the end of the waking period in the evening. Yet the molecular and circuit mechanisms by which the circadian clock modulates sleep homeostasis remain unclear.

To understand the mechanistic basis of circadian regulation of sleep homeostasis, we are using *Drosophila*, a well-established model for investigating the molecular and neural basis of circadian rhythms and sleep. Sleep is characterized by quiescence, increased arousal thresholds, changes in neuronal activity, and circadian and homeostatic regulation (*Campbell and Tobler, 1984*). Flies display each of these hallmarks (*Hendricks et al., 2000*; *Shaw et al., 2000*; *van Alphen et al., 2013*) and have simple, well-characterized circadian and sleep neural networks (*Dubowy and Sehgal, 2017*; *Shafer and Keene, 2021*). About 150 central pacemaker neurons that express molecular clocks (*Dubowy and Sehgal, 2017*). Of these, four small ventral lateral neurons (sLNvs) (per hemisphere) that express pigment dispersing factor (PDF) are necessary for driving morning activity in anticipation of lights on and exhibit peak levels of calcium around dawn (~ZT0) (*Grima et al., 2004*; *Liang et al., 2019*; *Liang et al., 2017*; *Stoleru et al., 2004*). The dorsal lateral neurons (LNds) and a 5th PDF$^-$ sLNv are necessary for evening anticipation of lights off and show a corresponding evening calcium peak (ZT8-ZT10) (*Gossan et al., 2014*; *Grima et al., 2004*; *Liang et al., 2019*; *Liang et al., 2017*; *Stoleru et al., 2004*). The posterior DN1 (DN1ps) consist of glutamate-positive (Glu$^+$) subsets necessary for morning anticipation and Glu$^-$ necessary for evening anticipation under low light conditions (*Chatterjee et al., 2018*). Lateral posterior neurons (LPN) are not necessary for anticipation but are uniquely sensitive to temperature cycling (*Miyasako et al., 2007*). Specific pacemaker subsets have been linked to wake promotion (PDF$^+$ large LNv (*Chung et al., 2009*; *Parisky et al., 2008*; *Sheeba et al., 2008*), diuretic hormone 31 (DH31$^+$) DN1ps [*Kunst et al., 2014*]) and sleep promotion (Glu$^+$ DN1ps (*Guo et al., 2016*), Allostatin A$^+$ LPNs [*Ni et al., 2019*]), independently of their clock functions. How these neurons regulate homeostatic sleep drive itself remains unsettled.

Timed signaling from these clock neurons converges on the neuropil of the ellipsoid body (EB). The sLNvs and LNds appear to communicate to R5 EB neurons through an intermediate set of dopaminergic PPM3 neurons based largely on correlated calcium oscillations (*Liang et al., 2019*). The anterior projecting subset of DN1ps provide sleep promoting input to other EB neurons (R2/R4M) via tubercular bulbar (TuBu) interneurons (*Guo et al., 2018*; *Lamaze et al., 2018*). Activation of a subset of these TuBu neurons synchronizes the activity of the R5 neurons which is important for sleep maintenance (*Raccuglia et al., 2019*). Critically, the R5 neurons are at the core of sleep homeostasis in *Drosophila* (*Liu et al., 2016*). R5 neuronal activity is both necessary and sufficient for sleep rebound (*Liu et al., 2016*). Extended sleep deprivation (12–24 hr) elevates calcium, the critical presynaptic protein BRUCHPILOT (BRP), and action potential firing rates in R5 neurons. The changes in BRP in this region not only reflect increased sleep drive following SD but also knockdown (KD) of *brp* in R5 decreases rebound (*Huang et al., 2020*) suggesting it functions directly in regulating sleep homeostasis. R5 neurons stimulate downstream neurons in the dorsal fan-shaped body (dFB), which are sufficient to produce sleep (*Donlea et al., 2014*; *Donlea et al., 2011*; *Liu et al., 2016*). Yet how the activity of key clock neurons are integrated with signals from the R5 homeostat to determine sleep drive remains unclear.

Here, we dissect the link between the circadian and homeostatic drives by examining which clock neural circuits regulate sleep rebound at different times of day in *Drosophila*. Akin to the forced desynchrony protocols, we enforced wakefulness at different times of day and assessed sleep rebound. We exposed flies to 7 hr cycles of sleep deprivation and recovery, enabling assessment of homeostasis at every hour of the day. We found that rebound is suppressed in the evening in a *Clk*-dependent manner. We demonstrate that time-dependent rebound is mediated by specific subsets of pacemaker neurons, independently of their effects on locomotor activity. Moreover, homeostatic R5 EB

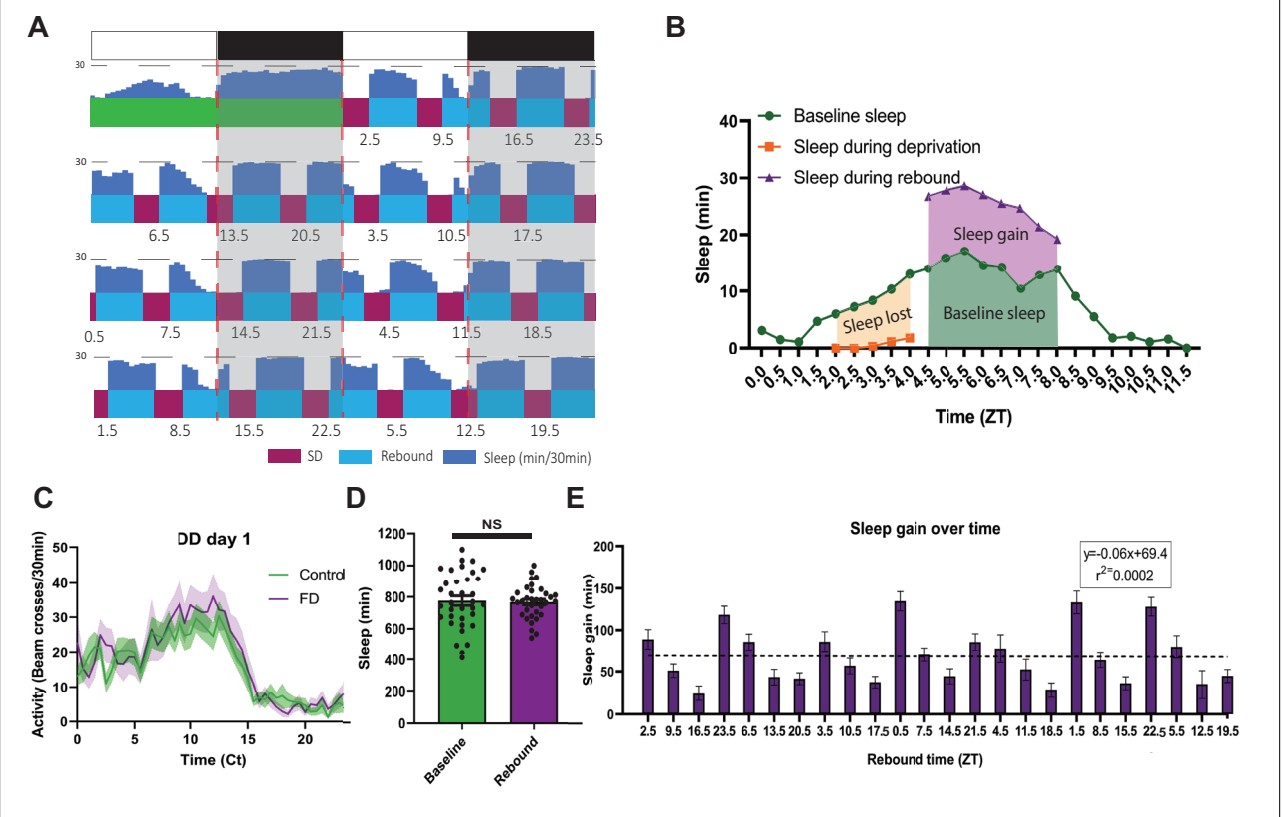

**Figure 1.** *Drosophila* forced desynchrony protocol can be used to illustrate time dependent rebound (**A**) Average WT sleep (N=32) over the final 8 days of SSD protocol with the time at which rebound begins (ZT) noted below each rebound period. (**B**) Profiles of sleep metrics used to compare rebound at different times of day (example is rebound occurring at ZT4.5). Sleep lost is determined by the difference between baseline sleep and sleep during the SD. Sleep gain is determined by the difference between rebound and baseline sleep. (**C**) Average activity of WT flies over 24 hr of flies released into the dark following SSD stimulation (N=19) or control (N=19) that received no stimulation. WT Flies released into DD1 following SSD display a profile of activity similar to control flies. Shaded bands indicate SEM.(**D**) Average sleep during baseline and the average sleep per day during the 7-day SD-rebound period (individual flies shown circles). There is no significant difference between average baseline sleep and average sleep per day over the course of the SSD (p>0.08, paired t-test). (**E**) Average WT (N=32) sleep gain across the course of the experiment with rebound time (ZT) depicted on the x axis. Regression of WT sleep gain over the course of the experiment displays no significant trend (p>0.95 linear regression). Data are means +/- SEM.

The online version of this article includes the following source data for figure 1:

**Source data 1.** T7 *Drosophila* forced desynchrony protocol can be used to illustrate time dependent rebound.

neurons integrate circadian timing and homeostatic drive; we demonstrate that activity dependent and presynaptic gene expression, BRP expression, neuronal output, and wake sensitive calcium levels are all elevated in the morning compared to the evening, providing an underlying mechanism for clock programming of time-of-day dependent homeostasis.

## Results

### Scheduled sleep deprivation demonstrates suppression of rebound in the evening

To confirm and resolve the timing of clock modulation of sleep rebound, we scheduled sleep deprivation in flies at different times of day and assessed sleep rebound, a protocol we term scheduled sleep deprivation (SSD). We employed an ultradian 7 hr cycle over 7 days allowing us to observe rebound at each hour of the 24 hr day (24 total deprivations) (*Figure 1a and b*). SD was administered for 2.5 hr followed by 4.5 hr of rebound such that flies would be allowed ~⅔ of the day to sleep, similar to the ratio of sleep observed in a WT female fly without SD. Given the potential for stress effects of longer deprivation typically used in flies (6–24 hr) we opted for a shorter 2.5 hr protocol. To test if SSD modulated the circadian phase, SSD flies released into constant dark (DD) following the protocol did not

exhibit any detectable change in phase (*Figure 1c*). There was no significant difference between total sleep in flies kept in SSD and those under baseline conditions (*Figure 1d*). In addition, sleep rebound does not increase over the course of the 7-day protocol further suggesting that flies are able to fully recover sleep during the 4.5 hr rebound period (*Figure 1e*). Together these results demonstrate that the SSD protocol allows assessment of rebound at different times of day without altering total sleep or circadian phase.

By comparing flies' baseline sleep to their rebound sleep (sleep after deprivation) around the clock, we observed robust rebound in the morning and suppressed rebound in the evening (*Figure 2a*). Under baseline conditions, flies typically show morning and evening peaks in wakefulness/activity. After sleep deprivation, flies display a robust sleep rebound throughout the 4.5 hr rebound period in the morning while evening rebound is suppressed (*Figure 2a*). To statistically compare morning and evening times of day here and throughout this study, we selected specific time points where the amount of sleep deprived and the baseline sleep during the rebound, two potential confounds, were comparable, allowing a direct comparison of sleep rebound. We also include standardized time points (ZTs 1.5 and 9.5) in the figure supplements for within time point comparisons. As indicated in the heat map, we found sleep rebound in the morning is significantly higher than sleep rebound in the evening when controlling for baseline sleep such that there is a>2 x difference in rebound between morning and evening time points (rebound at ZT1.5~133 min and ZT9.5~51 min) (*Figure 2b*). This was also accompanied by a significant difference in latency following deprivation (*Figure 2—figure supplement 1d*). We observed similar results using a streamlined protocol focusing on morning (ZT1.5 and 2.5) and evening timepoints (ZT8.5, 9.5, 10.5) (*Figure 2—figure supplement 1a*). During the course of our experiments, we transitioned to a more streamlined protocol to reduce the length of the protocol and the number of sleep deprivations, minimizing the potential for trends in sleep over the course of the protocol. Video evidence confirms that these morning/evening differences are not due to failure to cross the infrared beam due to increased feeding (*Videos 1 and 2*). Lastly, we determined if these effects persist under constant darkness (DD). We observed elevated rebound in the morning (CT2.5) relative to the evening (CT10.5), indicating that these differences are not dependent on light (*Figure 2c*). Altogether, these data suggest that homeostatic rebound sleep is strongly modulated by the internal clock.

## Sleep rebound is dependent on the molecular clock

To determine if morning/evening differences in rebound are due to the circadian clock we performed SSD in arrhythmic *Clk*^out (*Lee et al., 2014*) and short-period *per*^s mutants, which have an advanced evening peak in LD (*Hamblen-Coyle et al., 1992*; *Konopka and Benzer, 1971*). In the absence of *Clk*, flies do not display the wild-type morning and evening peaks of wakefulness and exhibit robust rebound at all times, reaching maximal levels of sleep after each SD (*Figure 2b*). Selected morning/evening time points do not exhibit significant differences in rebound in LD (ZT1.5 and ZT9.5) nor in DD (CT2.5 and CT10.5) (*Figure 2e and f*). There was also no difference in latency between baseline sleep matched morning and evening time points (ZT1.5 and ZT8.5) after sleep deprivation in *Clk*^out (*Figure 2—figure supplement 1e*). Similar to wild-type flies, *per*^s showed elevated rebound in the morning compared to the evening; however, as expected, the trough of rebound sleep in the evening was phase advanced relative to wild-type by about 4 hr (ZT5.5 v. ZT9.5) (*Figure 2—figure supplement 1b,c*). Furthermore, *per*^s flies exhibit an increased sleep latency following deprivation in earlier evening time points (ZT7.5) relative to control (ZT9.5) (*Figure 2—figure supplement 1c*). The loss of a morning/evening difference in rebound in arrhythmic *Clk*^out and the phase advance of evening rebound suppression in *per*^s further support the role of the clock in regulating sleep rebound.

## Glutamatergic DN1p circadian pacemaker neurons mediate morning and evening differences in rebound

To address the underlying neuronal basis, we employed a 'loss-of-function' approach where we inactivated and/or ablated targeted neuronal populations and assessed the impact on sleep rebound at different times of day. To test the role of clock neurons, we selectively ablated subsets by expressing the pro-apoptotic gene *head involution defective* (*hid*) using the Gal4/UAS system.

Ablation of most of the pacemaker neurons including those underlying morning and evening behavior using *cry39-Gal4* (*Klarsfeld et al., 2004*; *Picot et al., 2007*) substantially reduced both

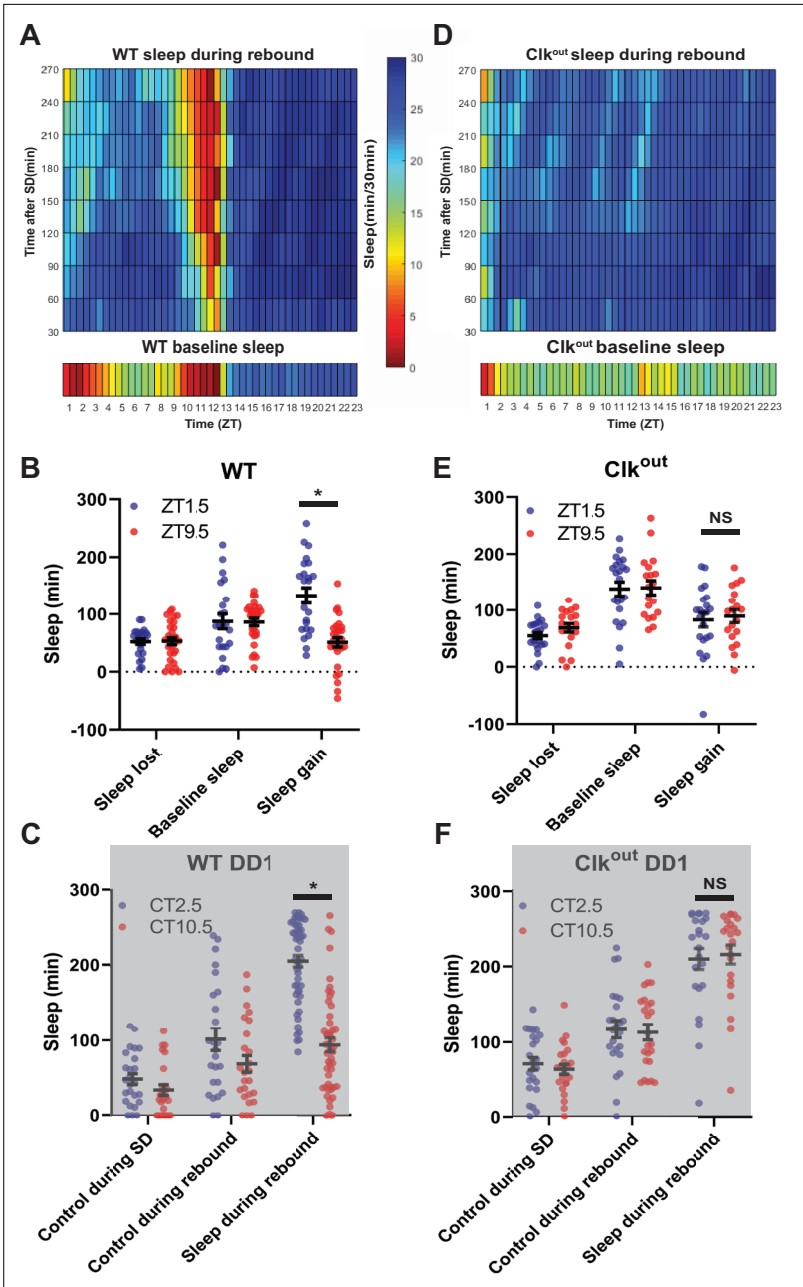

**Figure 2.** Sleep rebound is dependent on the molecular clock(A,D) Rebound sleep heatmaps (above) illustrate average sleep as a function of time of day when rebound occurred (ZT) and minutes after SSD episode. Missing time points are filled using matlab linear interpolation function. Baseline sleep heatmaps (below) illustrate average sleep during 30 min bins. (**A**) WT (N=32) baseline displays low sleep following lights on and preceding lights off. Immediately following SD flies show high sleep except in the hours preceding lights off. Flies tend to sleep less as rebound time proceeds. (**B,E**) Comparison of sleep lost, baseline sleep, and sleep gain following deprivation at morning and evening timepoints. (**B**) Sleep gain is greater for WT (N=32) rebound at ZT1.5 compared to ZT9.5 (p<0.00001, paired t-test). (**C,F**) Two sleep measures in control flies (control during SD and control during rebound), along with sleep during rebound in SD with rebound at 2.5 and 10.5. (**C**) Rebound sleep is greater following deprivation at CT2.5 compared to CT10.5 (p<0.00001, paired t-test) in WT flies (N=49). (**D**) $Clk^{out}$ (N=40) baseline sleep (below) is nearly constant except for low sleep immediately following lights on. SD uniformly increases sleep and flies tend to sleep less as rebound time proceeds. (**E**) No difference between sleep gain at the two time points is observed in $Clk^{out}$ (N=40) (p>0.37, paired t-test). (**F**) No difference in rebound sleep is observed in $Clk^{out}$ (N=23) (p>0.75, paired t-test). Data are means +/- SEM.

*Figure 2 continued on next page*

*Figure 2 continued*

The online version of this article includes the following source data and figure supplement(s) for figure 2:

**Source data 1.** Sleep rebound is dependent on the molecular clock.

**Figure supplement 1.** Total sleep and sleep latency vary as a function of time and SD (**A**) Comparison of sleep lost, baseline sleep, and sleep gain at morning (ZT1.5) and evening (ZT9.5) time points using abridged protocol with WT flies.

**Figure supplement 1—source data 1.** Total sleep and sleep latency vary as a function of time and SD.

morning and evening anticipation in males (*Table 1*) as previously described (*Stoleru et al., 2004*). Anticipation in females is more difficult to quantify due to more consolidated sleep and wake, that is, sleep at night reduces morning anticipation, more mid-day wake reduces evening anticipation (*Isaac et al., 2010*). Consistent with the loss of circadian function, ablation also abolished the difference between baseline sleep matched morning and evening rebound (*Figure 3a and b*), displaying high rebound across time points (*Figure 3—figure supplement 1a,b*). This effect was also observed using standardized morning and evening (ZT1.5\9.5) time points in which baseline sleep was not matched (*Figure 3*). This effect appears to be predominantly due to elevated rebound in the evening (*Figure 3—figure supplement 1e*). We ablated PDF⁺ neurons using *pdf-Gal4*, which we verified by observing substantially reduced morning anticipation in males validating our reagent (*Table 1*). Nonetheless we still observed substantially higher rebound in the morning (ZT1.5) versus the evening (ZT9.5) (*Figure 3c*). Moreover the examination of rebound using our full SSD does not indicate a clear change in phase that could explain these results (*Figure 3—figure supplement 1c*). Coupling *cry39-Gal4* with *pdf-Gal80* to ablate most clock cells except PDF⁺ neurons confirms this observation; these flies display high rebound across time points (*Figure 3—figure supplement 1d*) and comparably high rebound in baseline sleep matched morning and evening time points similar to *cry39-Gal4* (*Figure 3d*), highlighting the role of non-PDF clock neurons.

Potential synaptic targets of the PDF⁺ sLNv that are also important for morning behavior are the Glu⁺ DN1p neurons (*Chatterjee et al., 2018*; *Zhang et al., 2010a*; *Zhang et al., 2010b*). Targeting of the Glu⁺ DN1p has relied on drivers that are expressed outside of the DN1p including other sleep regulatory neurons (*Chatterjee et al., 2018*; *Guo et al., 2016*). To more definitively test their function, we employed the intersectional split Gal4 system (*Dionne et al., 2018*) utilizing two promoters, *R18H11* (expressed in DN1p and other neurons) (*Guo et al., 2016*) and *R51H05* that uses the vesicular glutamate transporter (vGlut) promoter, presumably targeting glutamatergic neurons. This intersection resulted in expression in just 6–7 neurons per hemisphere with little or no expression elsewhere in the brain. Selective labeling of dendritic and axonal arbors using DenMark (*Nicolaï et al., 2010*) and synaptotagmin GFP (*Zhang et al., 2002*), respectively, demonstrated that these neurons show presynaptic projections to both the pars intercerebralis(PI) and more modestly to the lateral posterior neuropil (*Figure 4a–c*), the latter consistent with a previous report (*Lamaze et al., 2018*). We

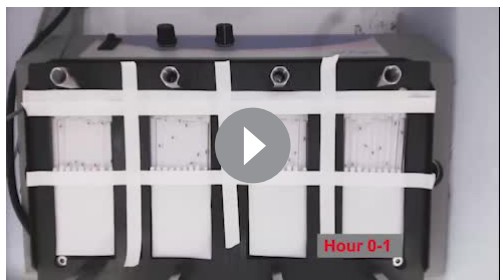

**Video 1.** Flies exhibit sleep following 2.5 hr SD terminating at ZT1.5 Sped up video recording of 4.5 hr of rebound of 36 WT flies following SD from ZT23-ZT1.5. Hours post SD are indicated in red in the bottom right corner. Flies exhibit little movement throughout the 4.5 hr following SD indicating sleep.

https://elifesciences.org/articles/74327/figures#video1

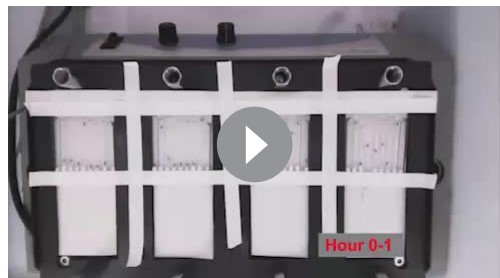

**Video 2.** Flies are active following 2.5 hr SD terminating at ZT9.5 Sped up video recording of 4.5 hr of rebound of 36 WT flies following SD from ZT7-ZT9.5. Hours post SD are indicated in red in the bottom right corner. After a brief period of immobility flies exhibit high activity (low sleep) preceding lights on.

https://elifesciences.org/articles/74327/figures#video2

**Table 1.** Summary of male morning and evening anticipation.

| Genotype | Region/Cells targeted | LD morning anticipation | LD evening anticipation | N |
|---|---|---|---|---|
| +>hid | Clock Gal4 control | 0.14+/-0.04 | 0.37+/-0.03 | 17 |
| pBDP split >hid | Split control (HID) | 0.13+/-0.02 | 0.24+/-0.03 | 26 |
| pBDP split >kir | Split control (Kir) | 0.10+/-0.02 | 0.33+/-0.04 | 12 |
| cry39>hid | broad clock | 0.05+/-0.02 ** | 0.12+/-0.04 *** | 30 |
| pdf >hid | PDF | –0.07+/-0.02 *** | 0.24+/-0.03* | 38 |
| cry39; pdf-gal80 >hid | LNd and Dn1 | 0.05+/-0.01 * | 0.06+/-0.04 *** | 14 |
| R51H05 AD; R18H11 DBD >hid | Glu +DN1 p | 0.06+/-0.02 * | 0.25+/-0.02 | 22 |
| MB122 >hid | 3–4 LNds PDF-sLNv | 0.12+/-0.02 | 0.25+/-0.02 | 35 |
| MB122 >kir | 3–4 LNds PDF-sLNv | 0.07+/-0.02 | 0.22+/-0.02 | 26 |

Data are means +/- SEM (*p<0.05, **p<0.01, ***p<0.001).

targeted *hid* expression using this split Gal4, we observed a reduction in morning anticipation in males demonstrating the necessity of this defined neuronal group (*Table 1*). However, in females used in our protocols, we did not observe a reduction in morning anticipation, possibly due to the lights-on activity peak masking anticipation (*Figure 4d–f*). We also did not observe significant changes in baseline sleep levels (*Figure 4g*). Despite the lack of a significant change in their baseline sleep/ activity profiles, ablation suppressed the difference in morning and evening rebound observed in both Gal4 and hid controls, although there was still a trend towards a morning-evening difference (*Figure 4h–j*). This effect was also observed using standardized morning and evening (ZT1.5\9.5) time points in which baseline sleep was not matched (*Figure 4—figure supplement 1c,d*). There was a significant difference in sleep gain at ZT1.5 between *hid* control flies and Glu⁺ DN1p ablated flies but did not reach significance with the Gal4 controls (*Figure 4i*). Overall, this indicates that Glu⁺ DN1ps may mediate differences between morning and evening rebound largely independent of their role in regulating baseline sleep/activity.

## TuBu and R2/R4m neurons are important for time-dependent modulation of sleep homeostasis

A subset of DN1ps send anterior projections to TuBu interneurons which in turn target the R2/R4m neurons of the EB (*Guo et al., 2018*; *Lamaze et al., 2018*; *Figures 4a and 5a*). TuBu neurons are a heterogeneous group distinguished by their axonal projections to 3 regions (superior, anterior and inferior) of the Bulb (BU), a neuropil comprised of, among other things, dendritic projections of neurons that form the EB (*Lovick et al., 2017*; *Omoto et al., 2017*). Previous studies have highlighted the role of the superior projecting TuBu neurons in generating sleep (*Guo et al., 2018*; *Lamaze et al., 2018*). To validate and further resolve this circuitry, we mined the Janelia Farm connectome which uses a large-scale reconstruction of the central brain from electron microscopy data (*Scheffer et al., 2020*). Using this approach, we identified direct synaptic connections from a subset of DN1pB (body IDs: 386834269, 5813071319) to a subset of TuBu neurons (TuBu01), to R4m neurons and eventually to R2 neurons (*Figure 5—figure supplement 1a,b*). Based on their morphology the TuBu01 neurons are anterior\inferior projecting. Thus, this connectome analysis both validated this circuit but also provided higher resolution for specific subsets that may be involved.

To determine if these neurons are important for sleep homeostasis, we first tested Gal4 drivers previously used to mark these neurons (*Guo et al., 2018*; *Lamaze et al., 2018*; *Liang et al., 2019*; *Liu et al., 2016*) in combination with *hid*, but found that in many cases (*R52B02*, *R20D01*) they were lethal, likely due to broader anatomic and/or developmental expression. So instead, we used the inward rectifying potassium channel *Kir2.1* (*Baines et al., 2001*) to silence these neurons and examined sleep rebound in the morning and evening. Silencing of a previously used driver (*R92H07*) that

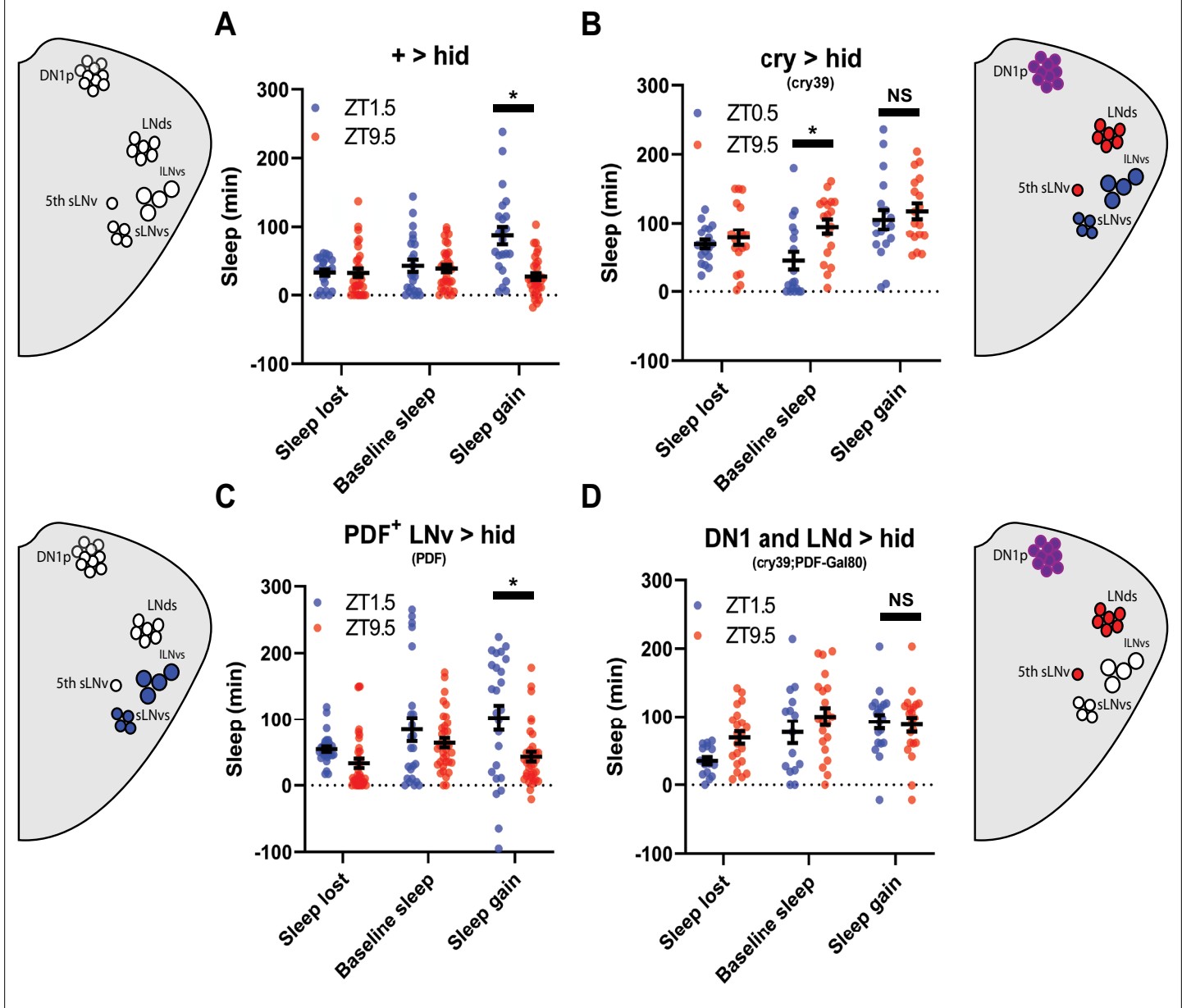

**Figure 3.** PDF+ neurons do not mediate morning/evening differences in rebound (**A,B,C,D**) Comparison of sleep lost, baseline sleep, and sleep gain following deprivation at morning and evening timepoints in clock neuron-ablated flies. Morning times are matched with evening time points with similar baselines. (**A**) Control flies with no ablated neurons (+>hid) (N=27) exhibit greater rebound in the morning compared to matched evening time point (p<0.0001, paired t-test). (**B**) Flies with most clock neurons ablated (cry39 >hid) (N=19) exhibit no difference in sleep gain between matched morning/evening time points (p>0.70, paired t-test). (**C**) Files with PDF+ neurons ablated (pdf >hid) (N=35) exhibit greater rebound in the morning compared to a matched evening time point (p<0.01, paired t-test). (**D**) Flies with most clock neurons ablated except PDF+ neurons (cry39; pdf-Gal80 >hid) (N=22) exhibit no significant difference in sleep gain between matched morning/evening time points (p>0.97, paired t-test). Data are means +/- SEM.

The online version of this article includes the following source data and figure supplement(s) for figure 3:

**Source data 1.** PDF+ neurons do not mediate morning/evening differences in rebound.

**Figure supplement 1.** Non PDF neurons drive morning/evening differences in rebound(A,B,D,E) Rebound sleep heatmaps (above) illustrate average sleep as a function of time of day when rebound occurred (ZT) and minutes after SSD episode.

**Figure supplement 1—source data 1.** Non PDF neurons drive morning/evening differences in rebound.

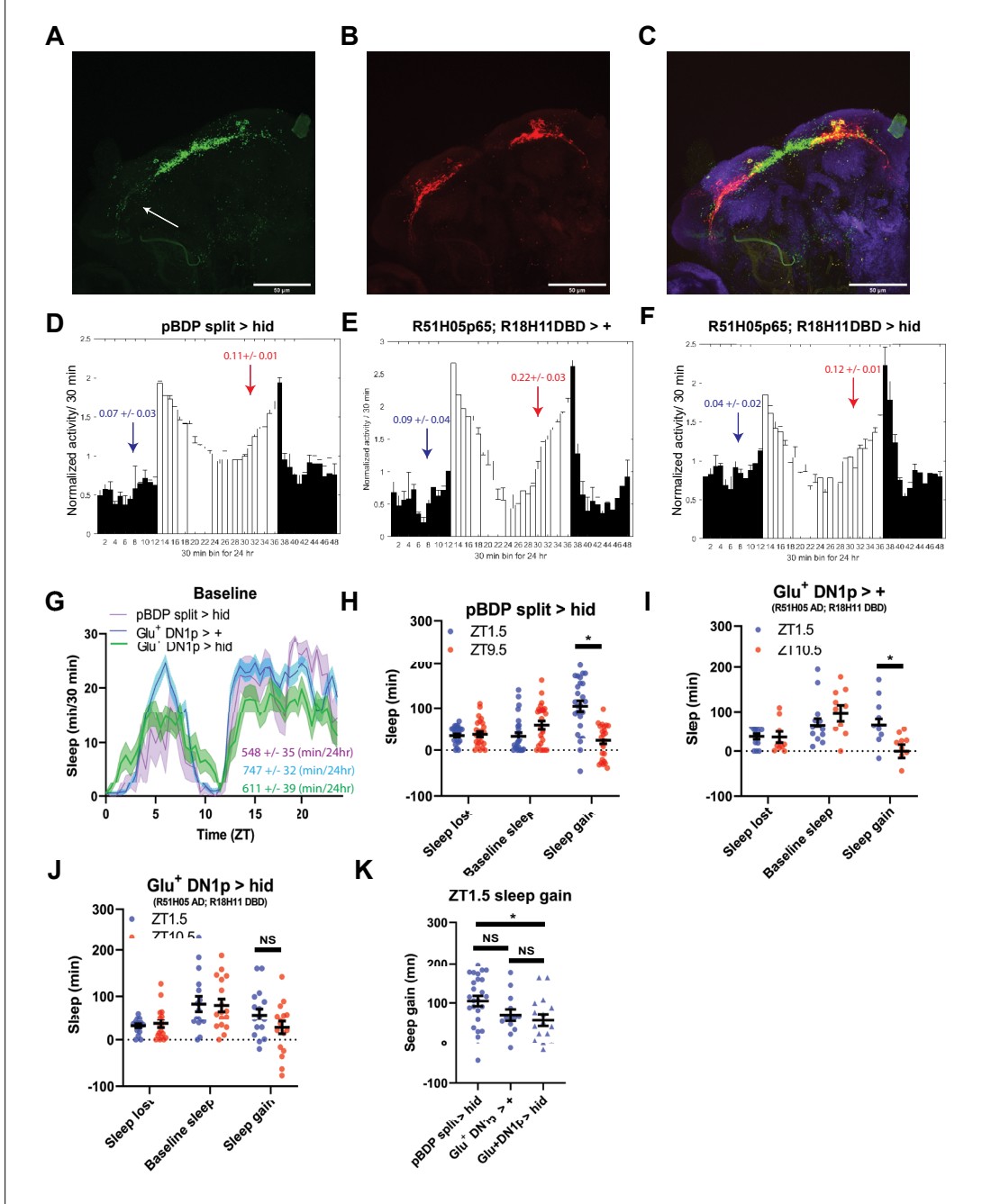

**Figure 4.** Glutamatergic DN1ps are necessary for morning and evening differences in rebound. (**A-C**) 20 x images of split Gal4 line that labeling presynaptic (**A**), postsynaptic (**B**) and overlay (**C**) of Glu+ DN1ps (R51H05 AD; R18H11 DBD >SYT GFP; DenMark) co-stained for BRP (blue). (**D–F**) Averaged activity eductions for female flies during the first 2 days of 12:12 LD. The light-phase is indicated by white bars while the dark-phase is indicated by black bars. Morning and evening anticipation indices are represented in blue and red respectively. (**G**) Average sleep during the baseline day for Glu+ DN1ps ablated (R51H05 AD; R18H11 DBD >hid) (N=30) (green), Gal4 control (R51H05 AD; R18H11 DBD> +) (N=36) and hid control (pBDP split >hid) (N=26) (purple). Sleep per 24 hr is indicated in the bottom right. (**H–J**) Comparison of sleep lost, baseline sleep, and sleep gain following deprivation at morning and evening timepoints in Glu+ DN1p ablated flies. Morning times are matched with evening time points with similar baselines. (**H**) hid control flies with no ablated neurons (pBDP split >hid) (N=26) exhibit greater rebound in the morning compared to matched evening time point (p<0.0001, paired t-test). (**I**) Gal4 control flies with no ablated neurons (R51H05 AD; R18H11 DBD> +) (N=19) exhibit greater rebound in the morning compared to matched evening time point (p<0.01, paired t-test) (**J**) Flies with Glu+ DN1ps ablated (R51H05 AD; R18H11 DBD >hid) (N=21) do not exhibit a significant difference in sleep gain between matched morning/evening time points (p>0.09, paired t-test). (**K**) Comparison of sleep gain at ZT1.5 between flies with Flies with Glu+ DN1ps ablated (R51H05 AD; R18H11 DBD >hid) (N=21) and their controls (pBDP split >hid) (N=26) and (R51H05

*Figure 4 continued on next page*

*Figure 4 continued*

AD; R18H11 DBD> +) (N=19). R51H05 AD; R18H11 DBD >hid flies exhibit significantly less rebound at ZT1.5 compared to hid control (p<0.05, ANOVA) and a non-significant decrease compared to Gal4 control (p>0.05, ANOVA). Data are means +/- SEM.

The online version of this article includes the following source data and figure supplement(s) for figure 4:

**Source data 1.** Glutamatergic DN1ps are necessary for morning and evening differences in rebound.

**Figure supplement 1.** Standardized time points show similar effects to points with matched baseline sleep (**A–B**) Rebound sleep heatmaps (above) illustrate average sleep as a function of time of day when rebound occurred (ZT) and minutes after SSD episode.

**Figure supplement 1—source data 1.** Standardized time points show similar effects to points with matched baseline sleep.

labels superior projecting TuBu neurons had no effect on rebound (*Figure 5—figure supplement 1c,d*). We identified another GAL4 driver (*R52B02*) that labels the superior and anterior and/or inferior subgroups previously implicated in sleep regulation (*Guo et al., 2018*; *Jenett et al., 2012*; *Lamaze et al., 2018*). We used this line in combination with *Kir2.1* and found that the difference between morning and evening rebound was lost, similar to what was observed after Glu⁺ DN1p ablation (*Figure 5b, c*). We knocked down the expression of a metabotropic glutamate receptor (mGluR) in these neurons using RNAi (*Guo et al., 2016*) and observed phenotypes very similar to silencing them (*Figure 5d and e*). To determine which neurons are acting downstream of TuBu, we targeted the R2/R4m neurons using *R20D01* (*Lamaze et al., 2018*). Silencing these neurons with *Kir2.1* eliminated the difference in rebound between baseline sleep matched morning and evening time points, phenocopying TuBu silencing (*Figure 5f*). This effect was also observed using standardized ZT1.5/9.5 morning/evening time points in which baseline sleep was not matched (*Figure 5—figure supplement 2a-c*). Taken together, these results demonstrate a role for the DN1p-Tubu-R2/R4m circuit in regulating time-dependent sleep rebound.

## PDF⁻ sLNv and LNds mediate evening suppression of sleep rebound

To determine the cellular basis of the evening rebound phenotype, we selectively ablated 2–3 LNds and the 5th sLNᵥ (4 neurons) using the highly specific MB122B split Gal4 line (*Guo et al., 2017*). This manipulation resulted in high rebound across time points in SSD (*Figure 6—figure supplement 1a,b*) similar to what was observed in *Clk^out* mutants. Furthermore, the difference in sleep gain between baseline sleep matched morning and evening time points was abolished (*Figure 6a–c*). This effect appears to be due to a large (two- to sixfold) increase in rebound in the evening (*Figure 6d*), with a more modest(~1.5-fold) effect in the morning (*Figure 6e*).We observed similar results with Kir2.1 (*Figure 6—figure supplement 2a,b*). Surprisingly, we did not observe significant effects on anticipation (*Figure 6f–g*; *Figure 6—figure supplement 2c,d*) or baseline sleep levels by ablation (*Figure 6i*) or silencing (*Figure 6—figure supplement 2e*). Differences between these baseline anticipation results and previously observed silencing effects on sleep may be due the use of constitutive versus inducible silencing (*Guo et al., 2017*). Nonetheless, these results indicate that the effects on rebound are largely independent of baseline anticipation/sleep levels. Thus, just 4 PDF⁻ LNd/sLNv cells are essential for clock control of rebound with an especially strong suppressive effect in the evening.

## PPM3, R5 and dFB neuron synaptic output is required for intact sleep homeostasis

The PPM3 and R5 neurons have been implicated as downstream of the LNd (*Figure 7a*; *Liang et al., 2019*). To test the effects of PPM3 on sleep homeostasis we blocked synaptic transmission by expressing tetanus toxin (TNT; *Broadie et al., 1995*) using *R92G05-Gal4* (*Liang et al., 2019*) and a novel split GAL4 targeting R5 neurons (*R58H05 AD; R48H04 DBD*; *Figure 7b*) As LNd calcium oscillations are synchronized with those in the PPM3, we hypothesized that PPM3 silencing may phenocopy LNd ablation, increasing rebound in the evening. However, whie PPM3 silencing did reduce the difference in rebound between baseline sleep matched morning and the evening time points (*Figure 7c and d*), this effect appears to be due to dramatically reducing rebound in both the morning and evening (*Figure 7—figure supplement 2a,b*). Therefore, if homeostatic relevant LNd output is targeting PPM3 neurons, it is inhibiting rather than exciting these neurons. Blocking R5 synaptic output also reduced rebound in both morning and evening (*Figure 7—figure supplement 2a,b*), consistent with the role of these neurons in mediating rebound from 12 hr SD terminating in the

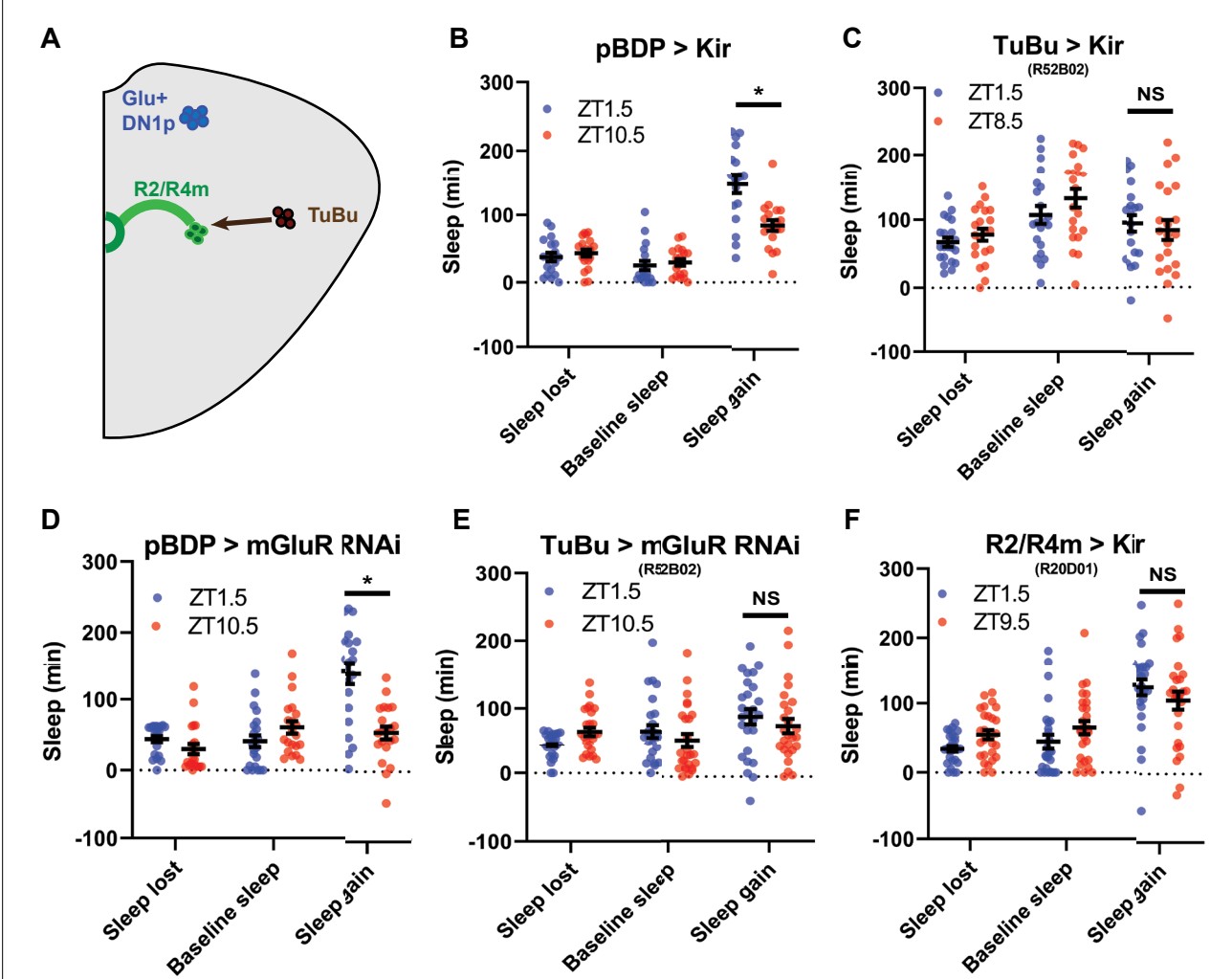

**Figure 5.** TuBu intermediates convey enhanced morning glutamatergic signal to R2/R4m ellipsoid body neurons (**A**) Cartoon illustrating proposed link between Glu+ DN1ps and R2/R4m with Tubu intermediates. (**B–F**) Comparison of sleep lost, baseline sleep, and sleep gain following deprivation at morning and evening timepoints while modulating neurons linking DN1ps to the EB. Morning times are matched with evening time points with similar baselines. (**B**) Enhancerless-Gal4 control flies (pBDP >Kir) (N=21) exhibit greater rebound in the morning compared to a matched evening time point (p<0.01, paired t-test). (**C**) Flies with TuBu neurons silenced (R52B02>Kir) (N=21) do not exhibit a difference in rebound between matched morning/evening time points (p>0.38, paired t-test). (**D**) Enhancerless-Gal4 driver paired with UAS-GluR-RNAi (pBDP >GluR RNAi) control (N=32) exhibit greater rebound in the morning compared to matched evening time point (p<0.00001, paired t-test). (**E**) Flies with KD of GluR in TuBu neurons (R52B02>GluR RNAi) do not exhibit a significant difference between matched morning/evening time points (p>0.28, paired t-test). (**F**) Flies with R2/R4m neurons silenced (R20D01>Kir) (N=32) do not exhibit a significant difference in rebound between matched morning/evening time points (p>0.26, paired t-test). Data are means +/- SEM.

The online version of this article includes the following source data and figure supplement(s) for figure 5:

**Source data 1.** TuBu intermediates convey enhanced morning glutamatergic signal to R2/R4m ellipsoid body neurons.

**Figure supplement 1.** Connectome analysis demonstrates link between anterior projecting DN1ps and R2/R4m ellipsoid body ring neurons.

**Figure supplement 1—source data 1.** Connectome analysis demonstrates link between anterior projecting DN1ps and R2/R4m ellipsoid body ring neurons.

**Figure supplement 2.** Standardized time points show similar effects to points with matched baseline sleep (**A–C**) Comparison of sleep lost, baseline sleep, and sleep gain following deprivation at morning and evening timepoints in clock neuron-ablated flies.

**Figure supplement 2—source data 1.** Standardized time points show similar effects to points with matched baseline sleep.

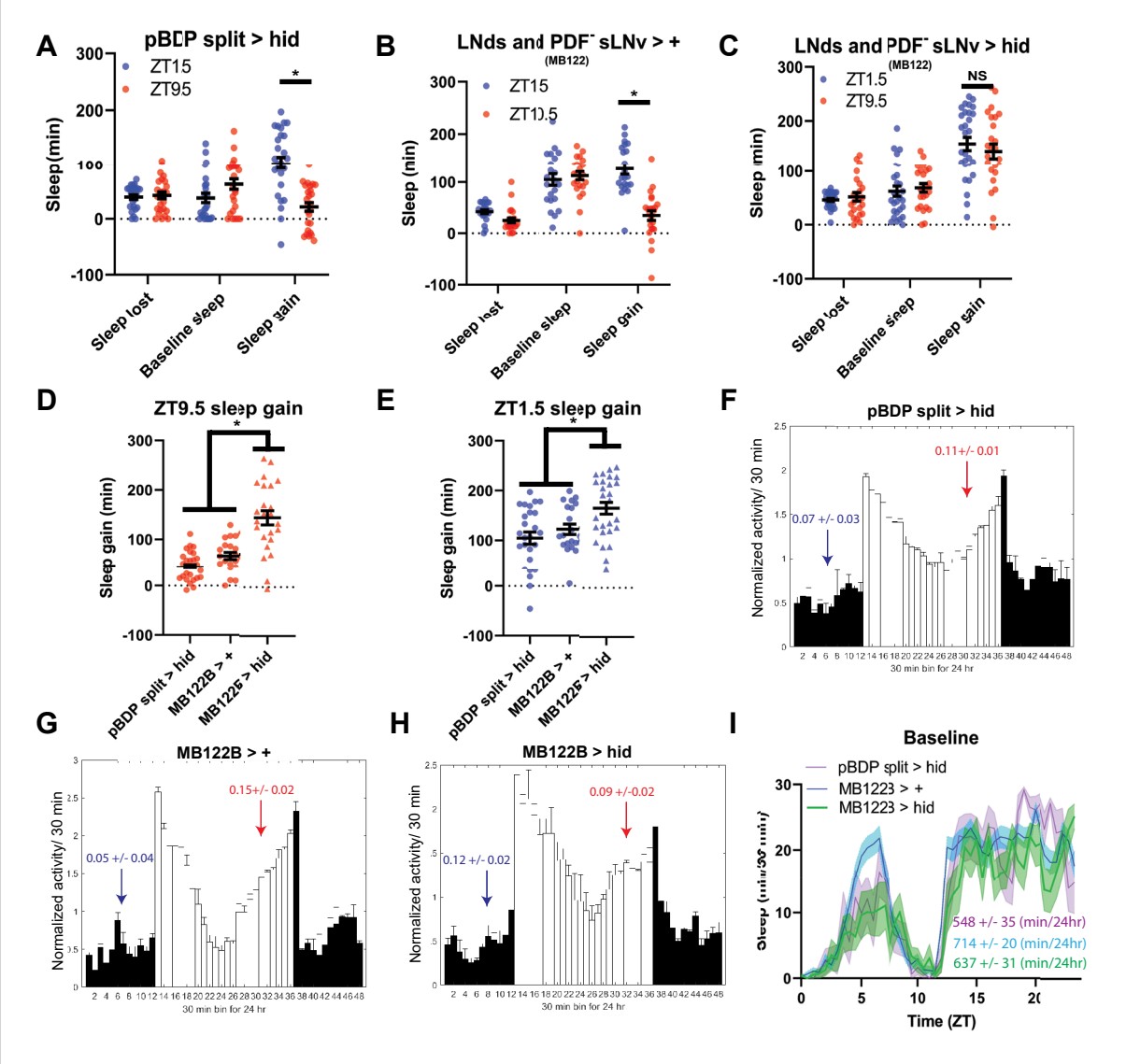

**Figure 6.** LNds and the PDF- sLNv suppress evening rebound (**A–C**) Comparison of sleep lost, baseline sleep, and sleep gain following deprivation at morning and evening timepoints in clock neuron-ablated flies. Morning times are matched with evening time points with similar baselines. (**A**) hid control flies with no ablated neurons (pBDP split >hid) (N=26) exhibit greater rebound in the morning compared to matched evening time point (p<0.0001, paired t-test). (**B**) Gal4 control flies with no ablated neurons (MB122B> +) (N=29) exhibit greater rebound in the morning compared to matched evening time point (p<0.01, paired t-test) (**C**) Flies with 2–3 LNds and the PDF- sLNv ablated (MB122B>hid) (N=30) do not exhibit a significant difference in sleep gain between matched morning/evening time points (p>0.50, paired t-test). (**D–F**) Averaged activity eductions for female flies during the first 2 days of 12:12 LD. Light phase is indicated by white bars while the dark phase is indicated by black bars. Morning and evening anticipation indices are represented in blue and red respectively. (**G**) Average sleep during the baseline day for LNds and the PDF- sLNv ablated (MB122B>hid) (N=30) (green), Gal4 control (MB122B > +) (N=29) (blue), and hid control (pBDP split >hid) (N=26) (purple). (**H, I**) Comparison of sleep gain at ZT1.5 (**H**) and ZT9.5 (**I**) between flies with 2–3 LNds and the PDF- sLNv ablated (MB122B>hid) (N=29) and their controls (pBDP split >hid) (N=26)(MB122B> +) (N=29). MB122B>hid flies exhibit greater rebound at morning (**H**) (p<0.05, ANOVA) and evening (**I**) (p<0.001, ANOVA). Data are means +/- SEM.

The online version of this article includes the following source data and figure supplement(s) for figure 6:

**Source data 1.** LNds and the PDF- sLNv suppress evening rebound.

**Figure supplement 1.** Full SSD demonstrates loss of evening suppression of rebound in LNd ablated flies (**A–B**) Rebound sleep heatmaps (above) illustrate average sleep as a function of time of day when rebound occurred (ZT) and minutes after SSD episode.

**Figure supplement 1—source data 1.** Full SSD demonstrates loss of evening suppression of rebound in LNd ablated flies.

**Figure supplement 2.** Silencing LNds and the PDF- sLNv suppresses evening rebound (**A–B**) Comparison of sleep lost, baseline sleep, and sleep gain following deprivation at morning and evening timepoints in clock neuron-ablated flies.

**Figure supplement 2—source data 1.** Silencing LNds and the PDF- sLNv suppresses evening rebound.

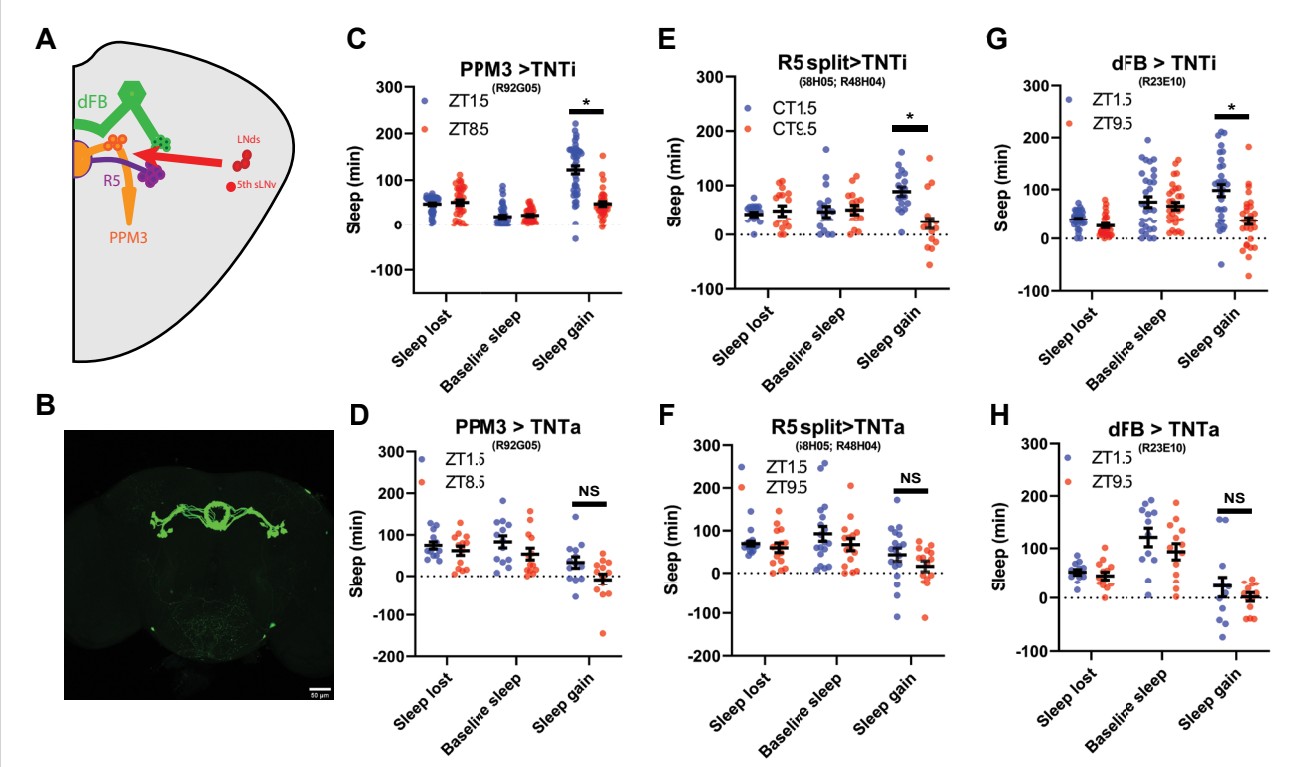

**Figure 7.** PPM3 convey enhancing homeostatic signal to R5 ellipsoid body neurons (**A**) Cartoon illustrating link between LNds and 5th sLNv and dFB via with PPM3 and R5 intermediates. (**B**) GFP Expression pattern of split Gal4 line that labels Glu⁺ DN1ps (R58H05 AD; R48H04 DBD >GFP) at 20 x. (**C–H**) Comparison of sleep lost, baseline sleep, and sleep gain following deprivation at morning and evening timepoints modulating neurons linking LNd activity to the EB. Morning times are matched with evening time points with similar baselines. (**C**) Flies expressing an inactive form of tetanus toxin in PPM3 neurons (R92G05>TNTi)(N=45) exhibit greater rebound in the morning than at a matched evening time point (p<0.0001, paired t-test). (**D**) Silencing PPM3 neurons with an active form of tetanus toxin (R92G05>TNTa)(N=27) resulted in no significant difference between matched morning/ evening time points (p>0.10, paired t-test). (**E**) Flies expressing an inactive form of tetanus toxin in R5 neurons (R58H05 AD; R48H04 DBD >TNTi) (N=21) exhibit greater rebound in the morning than at a matched evening time point (p<0.01, paired t-test). (**F**) Silencing R5 neurons with tetanus toxin (R58H05 AD; R48H04 DBD >TNTa) (N=16) resulted in no significant difference in sleep gain for matched morning and evening time points (p>0.70, paired t-test). (**G**) Flies expressing an inactive form of tetanus toxin in the dFB (R23E10>TNTi) (N=30) exhibit greater rebound in the morning than at a matched evening time point (p<0.0001, paired t-test). (**H**) Silencing dFB neurons with tetanus toxin (R23E10>TNTa) (N=12) resulted in no significant difference between morning and evening time points (p>0.45, paired t-test).

The online version of this article includes the following source data and figure supplement(s) for figure 7:

**Source data 1.** PPM3 convey enhancing homeostatic signal to R5 ellipsoid body neurons.

**Figure supplement 1.** Silencing of PPM3, dFB and R5 reduces rebound in the morning and evening (**A, B**) Comparison of sleep gain at morning (**A**) and evening (**B**) timepoints between flies with silenced neurons (TNTa) and their controls (TNTi).

**Figure supplement 1—source data 1.** Silencing of PPM3, dFB, and R5 reduces rebound in the morning and evening.

**Figure supplement 2.** Standardized time points show similar effects to points with matched baseline sleep (**A–C**) FD sleep during two baseline time periods (sleep lost and baseline sleep) and sleep gain for rebound occurring at ZT1.5 and ZT 9.5.

**Figure supplement 2—source data 1.** Standardized time points show similar effects to points with matched baseline sleep.

morning (*Liu et al., 2016*). Moreover, no difference between baseline sleep matched morning and evening rebound was evident (*Figure 7e and f*). R5 neurons promote sleep in response to deprivation by activating the sleep promoting dFB (*Liu et al., 2016*). Thus, we also blocked synaptic output from the dFB using TNT. Rebound at both morning and evening time points was reduced (*Figure 7— figure supplement 2a,b*) similar to what was reported for rebound beginning in the morning (*Qian et al., 2017*). This too resulted in no difference in rebound between baseline sleep matched morning/ evening time points as it was for PPM3 and R5 (*Figure 7g and h*). Although the exact nature of the PPM3 input remains an open question, these studies highlight a role for a PPM3-R5-dFB pathway in rebound sleep in response to deprivation at all times of day even with shorter deprivation protocols.

# R5 ellipsoid body neurons exhibit elevated expression of activity-dependent and presynaptic genes in the morning relative to the evening

To ascertain how the circadian system may impact the R5 homeostat, we examined molecular and physiological changes in R5 as a function of time and sleep need. Interestingly, activation and deprivation studies have focused exclusively on morning rebound. To identify time- and wake-dependent gene expression in an unbiased manner, we selectively labeled R5 neurons (*Figure 7b*, *R58H05 AD; R48H04 DBD >GFP*) and subjected flies to 2.5 hr of mechanical SD in either the morning or evening. We then isolated R5 neurons from control or SD flies at ZT1 and ZT9 using fluorescence-activated cell sorting and subjected them to RNA-sequencing.

Based on our behavioral data, we hypothesized that morning SD would induce differential gene expression compared to control flies that did not receive SD while evening SD would not be sufficient to induce changes in gene expression compared to controls. We were surprised to find that neither morning nor evening SD had much of an effect on gene expression in the R5 neurons (*Figure 8a and b*). In the morning, only two genes were significantly differentially expressed (q<0.1, *Hsp70Bb* and *stv*). Likewise, in the evening, only four genes were significantly differentially expressed (q<0.1, *CG5522*, *CG13285*, *mt:ND5*, and *Hsp70Bb*). In stark contrast, comparisons of morning and evening timepoints with or without sleep deprivation (Morning Control (MC) vs Evening Control (EC), Morning SD (MSD) vs Evening SD (ESD), or MC +MSD vs EC +ESD) produces 46–128 differentially expressed genes (q<0.1, *Figure 8c, d and e*). Notably, this time-of-day dependent regulation does not appear to be driven by core clock genes in these neurons (*Figure 8—figure supplement 1*). *Clk* is detected in only 2 out of 12 samples and only at very low levels in those samples with the expression of other clock genes like *per* and *tim* not fluctuating between the two timepoints.

To understand what sorts of molecular programs are undergoing differential regulation between morning and evening, we examined gene ontologies of genes upregulated in the morning. These terms include cellular components like 'presynaptic active zone', 'synaptic vesicle', 'terminal bouton', and 'cAMP-dependent protein kinase complex', as well as molecular functions like 'calcium ion binding' and 'calcium, potassium::sodium antiporter activity'. The genes identified in these categories suggest a temporally regulated state of activity for the R5 neurons. Indeed, major active zone regulators such as *Syx1A*, *Rim*, *unc-104*, *Srpk79D*, and *nSyb* are all significantly upregulated in the morning (*Figure 8e and f*). *Syx1A*, *Rim*, and *nSyb* are part of the synaptic vesicle docking and exocytosis machinery and *Rim* also regulates the readily-releasable pool of synaptic vesicles, playing a major role in presynaptic homeostasis (*Broadie et al., 1995*; *Müller et al., 2012*). *unc-104* is involved in trafficking of synaptic vesicles and BRP to the active zone (*Zhang et al., 2017*) and the kinase *Srpk79D* regulates trafficking and deposition of BRP at active zones via phosphorylation of its N-terminus (*Johnson et al., 2009*; *Nieratschker et al., 2009*). We also observed significant upregulation of genes involved in ionic transport across the plasma membrane, including *para*, a voltage-gated sodium channel (*Catterall, 2000*; *Loughney et al., 1989*), and *CG5890*, a predicted potassium channel-interacting protein (KChIP) (*Figure 8e and g*). Mammalian KChIPs have been shown to interact with voltage-gated potassium channels, increasing current density and conductance and slowing inactivation (*An et al., 2000*). Two sodium:potassium/calcium antiporters, *CG1090* and *Nckx30C*, were also upregulated (*Figure 8e and g*). These antiporters function primarily in calcium homeostasis by using extracellular sodium and intracellular potassium gradients to pump intracellular calcium out of the cell when calcium levels are elevated (*Haug-Collet et al., 1999*). Amongst the most significantly upregulated genes in our dataset, we found six genes that were previously identified as activity-regulated genes in *Drosophila* (ARGs; *sr*, *Cdc7* (also known as *l(1)G0148*), *CG8910*, *CG14186*, *CG17778*, *hr38*) (*Figure 8e and h*). These genes are analogous to immediate early genes in mammals and represent half of a group of twelve genes that were induced in three distinct paradigms of neuronal stimulation (*Chen et al., 2016*). Finally, we found that several critical components of Creb signaling were enriched in the morning in R5 neurons (*Figure 8e and i*). *CrebA* was the most significantly upregulated gene in the morning samples, though we also saw significant increases in *meng*, which encodes a kinase that works synergistically with the catalytic subunits of PKA to phosphorylate and stabilize CREBB (*Lee et al., 2018*), as well as both regulatory subunits of PKA (*Pka-R1*, *Pka-R2*) (*Figure 8e and i*). CREBA and CREBB likely serve different roles, but appear to be involved in activity-dependent processes like dendritogenesis and long term memory (*Iyer et al., 2013*; *Yin et al., 1995*).

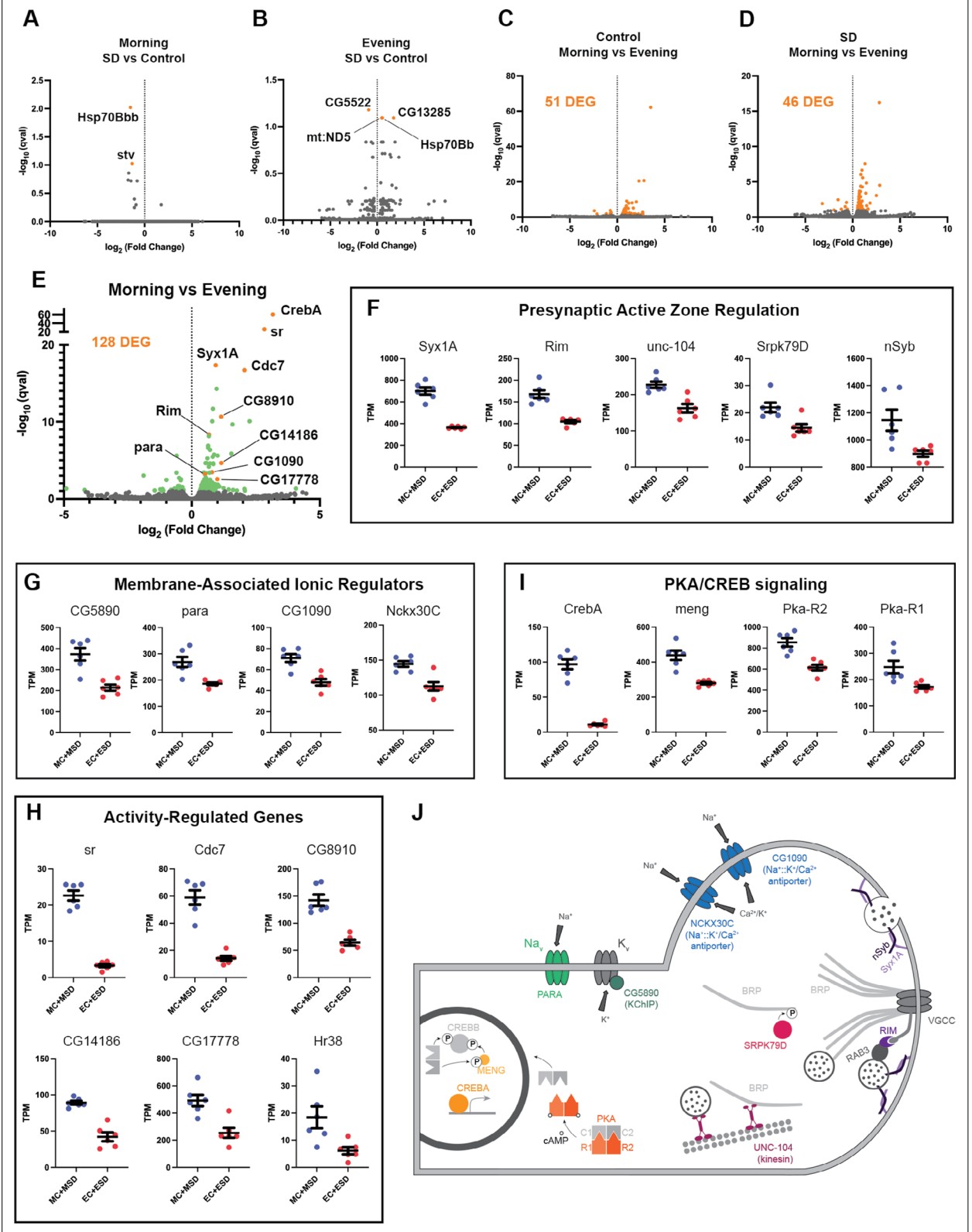

**Figure 8.** RNA sequencing of FAC-sorted R5 neurons suggests elevated activity in the morning (**A**) Volcano plot (fold change versus qval) of Morning SD (MSD) vs Morning Control (MC) gene expression. Significantly differentially expressed genes shown in orange. (**B**) Volcano plot of Evening SD (ESD) vs Evening Control (MC) gene expression. Significantly differentially expressed genes shown in orange. (**C**) Volcano plot of MC vs EC gene expression. 51 significantly differentially expressed genes (DEG) were identified and are shown in orange. (**D**) Volcano plot of MSD vs ESD gene expression. 46

*Figure 8 continued on next page*

*Figure 8 continued*

significantly differentially expressed genes (DEG) were identified and are shown in orange. (**E**) Volcano plot of MC +MSD vs EC +ESD gene expression. Differentially expressed genes are shown in green with a few genes highlighted in orange and labeled. (**F–I**) Scatter plots for several differentially expressed genes. Transcripts *Per Kilobase Million (TPM)* is shown for each sample. All morning samples are grouped, and all evening samples are grouped. Graphs are grouped by similar functions: (**F**) active zone components/regulators, (**G**) membrane-associated ionic regulators, (**H**) activity-regulated genes, (**I**) PKA/CREB signaling. (**J**) Schematic of select morning upregulated genes. Upregulated genes are shown in color while other interacting components are depicted in gray. PARA and CG5890 are both involved in the generation and propagation of action potentials. Multiple active zone components/regulators (NSYB, SYX1A, RIM, SRPK79D, UNC-104) interact with BRP and voltage-gated calcium channels (VGCCs) to support neuronal output and intracellular calcium influx. Elevated levels of intracellular calcium are regulated by the antiporters NCKX30C and CG1090. Second messenger cAMP interacts with regulatory subunits of PKA (R1/R2) and releases the catalytic subunits (C1/C2) to phosphorylate CREBB and MENG, stabilizing CREBB. CREBA acts as a transcriptional activator independent of PKA activity.

The online version of this article includes the following source data and figure supplement(s) for figure 8:

**Source data 1.** RNA sequencing of FAC-sorted R5 neurons suggests elevated activity in the morning.

**Figure supplement 1.** Clock genes are not different between morning and evening in R5 neurons Scatter plots for core clock genes.

**Figure supplement 1—source data 1.** Clock genes are not different between morning and evening in R5 neurons.

---

Synthesizing these data, it appears that a complex time-dependent program of transcriptional regulation is in play in the morning to upregulate the activity of R5 neurons (*Figure 8J*). Upregulation of *unc-104, Srpk79D, Syx1a, Rim*, and *nSyb* suggests that R5 neurons are assembling a greater number of mature active zones for neuronal output. Upregulation of *para* and the predicted KChIP *CG5890*, which should increase the voltage-gated conductance of sodium and potassium ions across the membrane, supports the idea that R5 neurons may be primed for greater action potentials in the morning. Upregulation of the two sodium:potassium/calcium antiporters suggests that intracellular calcium levels are elevated in the morning, again consistent with the idea that these neurons are more active in the morning. Significantly elevated levels of six ARGs also support this conclusion. Finally, there is some suggestion that the elevated activity may result in plasticity in the R5 neurons supported by PKA and CREB signaling.

---

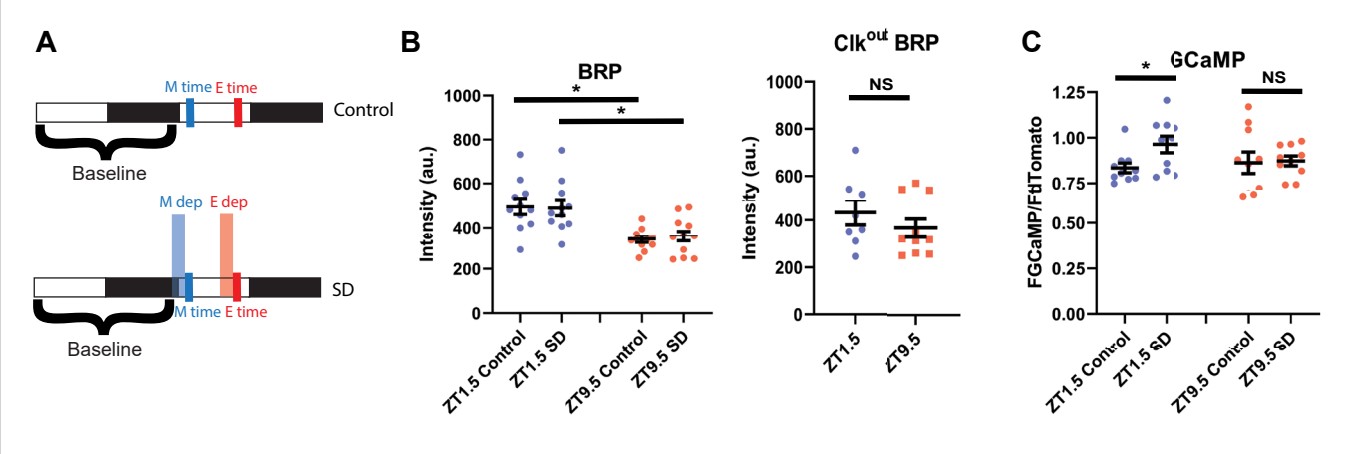

**Figure 9.** R5 neurons exhibit time dependent changes in BRP and calcium response to SD (**A**) Schematic illustrating deprivation and dissection timing for morning (**M**) and evening (**E**) with (lower) and without (upper) SD. (**B**) Fluorescence of BRP-STaR in R5 projections as a function of time of day and SD in WT (left)and *Clk^out* mutant (right) backgrounds. Intensity of BRP staining is decreased at ZT9.5 compared to ZT1.5 in both control (N=11, 11) (p<0.001, independent t-test) and SD (N=11, 11) (p<0.01, independent t-test) groups. Intensity of BRP staining is not affected by SD in the morning (N=11,11) (p>0.90, independent t-test) or evening (N=11,11) (p>0.58, independent t-test). Intensity of BRP staining in *Clk^out* mutants is not significantly different at ZT1.5 (N=8) compared to ZT9.5 (N=10) (p>0.36, independent t-test).(**D**) GCaMP expression in R5 projections (R69F08>GCamP6 s) at ZT1.5 and ZT9.5 with and without SD. GCaMP fluorescence was normalized to the tdTomato fluorescence signal intensity. There is no difference in normalized GCaMP6s signaling between baseline morning (N=10) and evening (N=10) time points. SD in the morning (N=10) increases GCaMP6s intensity (p<0.05, independent t-test) but not in the evening (N=10) (p>0.87 independent t-test), independent t-test. Data are means +/- SEM.

The online version of this article includes the following source data for figure 9:

**Source data 1.** R5 neurons exhibit time dependent changes in BRP and calcium response to SD.

## R5 neurons exhibit time-dependent changes in BRP and calcium response to SD

SD/extended wake results in the upregulation of many synaptic proteins (*Gilestro et al., 2009*). Most notable is the presynaptic scaffolding protein BRP, which is important for synaptic release (*Matkovic et al., 2013*), and is upregulated in the R5 neurons following 12 hr of SD (*Liu et al., 2016*). KD of *brp* in R5 neurons decreases rebound response to SD (*Huang et al., 2020*), suggesting that it is necessary for accumulating and/or communicating homeostatic drive. We hypothesized that differences in the propensity for R5 to induce sleep rebound in the morning/evening may be due to changes in synaptic strength that can be observed by tracking levels of BRP.

To test this idea, we used the synaptic tagging with recombination (STaR) system to selectively express a V5 epitope-tagged BRP in R5 neurons using the FLP/FRT system (*Chen et al., 2014*) as previously reported (*Liu et al., 2016*). We examined BRP at ZT1.5 and ZT9.5 with and without SD and found that BRP levels are higher at ZT1.5 than ZT 9.5 (*Figure 9a and b*). Interestingly, 2.5 hr SD had no effect on BRP intensity at either time point (*Figure 9b*). It is possible that BRP changes in response to 2.5 hr of SD are not observable, while a longer 12 hr deprivation is required to induce sufficient changes for observation (*Liu et al., 2016*). We next tested the same two time points in the *Clk^{out}* mutant background and found no significant difference between ZT1.5 and ZT9.5 (*Figure 9b*). As reduced BRP expression in the R5 reduces rebound (*Huang et al., 2020*), it is possible that clock-dependent changes in expression of BRP and associated presynaptic modifications are driving the difference in rebound observed in morning/evening.

The calcium concentration in R5 neurons increases following twelve hours of SD, suggesting that extended wakefulness can induce calcium signaling in these neurons. Blocking the induction of calcium greatly reduces rebound, supporting a critical role for calcium signaling in behavioral output (*Liu et al., 2016*). Furthermore, R5 neurons display morning and evening cell-dependent peaks in calcium activity across the course of the day indicating that calcium is also modulated by the clock network (*Liang et al., 2019*). It is unclear whether the circadian clock can modulate wake-dependent changes in calcium activity in the R5 neurons.

To test this idea, we expressed the calcium reporter GCaMP6s (*Chen et al., 2013*) in the R5 and examined calcium in the morning (ZT1.5) and evening (ZT9.5) with and without SD (*Figure 9a*). Interestingly there was no difference between the non-SD flies at each time point (*Figure 9c*). One study showed elevated calcium at a later evening timepoint (ZT12; *Liu et al., 2016*). Our finding of similar calcium levels may be because the morning time point resides on the downswing of the morning-peak of R5 calcium activity while the evening time point resides on the upswing of the evening calcium peak (*Liang et al., 2019*). Nonetheless, an SD induced increase in calcium was observed in the morning but suppressed in the evening (*Figure 9c*), suggesting that the R5 sensitivity to sleep deprivation is gated by the clock.

## Discussion

Here, we describe the neural circuit and molecular mechanisms by which discrete populations of the circadian clock network program the R5 sleep homeostat to control the homeostatic response to sleep loss. We developed a novel protocol to administer brief duration SD and robustly measure homeostatic rebound sleep. Using this strategy, we demonstrated that homeostatic rebound is significantly higher in the morning than in the evening. We then identified distinct subsets of the circadian clock network and their downstream neural targets that mediate the enhancement and suppression of morning and evening rebound respectively. Using unbiased transcriptomics, we observed very little gene expression significantly altered in response to our 2.5 hr sleep deprivation. On the other hand, we did identify elevated expression of activity-dependent and presynaptic genes in the morning independent of sleep deprivation. Consistent with this finding, we also observe elevated levels of the presynaptic protein BRP that is absent in the absence of *Clk*. These baseline changes are accompanied by an elevated calcium response to sleep deprivation in the morning mirroring the enhanced behavioral rebound in the morning. Taken together, our data support the model of a circadian regulated homeostat that turns the homeostat up late at night to sustain sleep and down late in the day to sustain wake.

Our studies suggest that homeostatic drive in the R5 neurons is stored post-transcriptionally. As part of our studies, we developed a novel protocol using minimal amounts of SD which could be useful for minimizing mechanical stress effects and isolating underlying molecular processes crucial for sleep homeostasis. Six to 24 hr of SD in *Drosophila* is commonly used despite the potential stressful or even lethal effects (*Fernandez et al., 2014*; *Shaw et al., 2002*; *Vaccaro et al., 2020*). Here, we demonstrate that shorter 2.5 hr deprivations not only induce a robust rebound sleep response (*Figure 2*), but also the percent of sleep lost recovered at ZT0 is close to 100% versus 14–35% seen in 12 hr SD protocols (*Blum et al., 2021*; *Kayser et al., 2014*; *Nall and Sehgal, 2013*; *Oh et al., 2014*). Using this shorter SD, we now find that many effects observed in R5 neurons with 12 hr SD (e.g. increased BRP and upregulation of *nmdar* subunits) are no longer observed with shorter SD, even though the necessity of R5 neurons for rebound is retained after 2.5 hr SD (*Figure 7e and f*). Previously, translating ribosome affinity purification (TRAP) was used to show upregulation of *nmdar* subunits following 12 hr SD (*Liu et al., 2016*).FACS and TRAP are distinct methodologies for targeted collection of RNA for sequencing and can yield unique gene lists (*Cedernaes et al., 2019*). One possibility is that upregulation of *nmdar* subunits is occurring locally in neuronal processes, which are often lost during FACS, and/or is at the level of translation initiation or elongation. Nonetheless, in agreement with previous work, we observed SD-induced increases in calcium correlated with behavioral rebound in the morning, suggesting that this process is a core feature of the cellular homeostatic response.

Using genetically targeted 'loss-of-function' manipulations, we have defined small subsets of circadian clock neurons and downstream circuits that are necessary for intact clock modulation of sleep homeostasis. The use of intersectional approaches enabled highly resolved targeting not possible with traditional lesioning experiments in the SCN (*Easton et al., 2004*). Collectively our studies defined a potential Glu⁺ DN1p-TuBu-R4m circuit important for enhancing morning rebound as well as a discrete group of LNds important for suppressing evening rebound. Importantly, most of these effects on sleep rebound are evident in the absence of substantial changes in baseline activity, despite other studies indicating their necessity for normal circadian behavior. Of note, the proposed roles of the DN1p and LNd clock neurons are sleep (*Guo et al., 2016*) and wake promotion (*Guo et al., 2018*) consistent with our findings after sleep deprivation. We hypothesize that by using chronic silencing methods, baseline effects may not be evident due to compensatory changes but that these effects are only revealed when the system is challenged by sleep deprivation. Similar genetic strategies in mammals (see *Collins et al., 2020*) may be useful in uncovering which SCN neurons are driving circadian regulation of sleep homeostasis given the comparable suppression of sleep rebound in the evening in humans (*Dijk and Czeisler, 1994*; *Dijk and Czeisler, 1995*; *Dijk and Duffy, 1999*; *Lazar et al., 2015*). Nonetheless, the finding of sleep homeostasis phenotypes in the absence of significant baseline effects suggests that a major role of these clock neuron subsets may be to manage homeostatic responses.

Our studies suggest that circadian and homeostatic processes do not compete for influence on a downstream neural target but that the circadian clock programs the homeostat itself. Using an unbiased transcriptomic approach, we discovered time-dependent expression of activity dependent and presynaptic genes (*Figure 8*), consistent with previous data that the R5 neurons exhibit time-dependent activity (*Liang et al., 2019*; *Liu et al., 2016*). We observed significant upregulation of several genes involved in synaptic transmission (*Syx1a, Rim, nSyb, unc-104, Srpk79D, para, CG5890*) evincing a permissive active state for R5 neurons in the morning. This is accompanied by elevated levels of the key presynaptic protein BRP in the morning compared to evening. It is notable that elevated BRP in the morning is the opposite of what would be expected based on a sleep-dependent reduction in BRP proposed by the synaptic homeostasis hypothesis (*Tononi and Cirelli, 2014*), suggesting a sleep-wake independent mechanism. Previous studies have shown that modulation of BRP levels in the R5 are important for its sleep function (*Huang et al., 2020*), suggesting that changes in BRP levels impact R5 function. We hypothesize that these baseline transcriptomic changes underlie the differential R5 sensitivity to sleep deprivation is evident as calcium increases in the morning and not the evening. Indeed, trancriptomic and proteomic studies of the mouse forebrain across time and after sleep deprivation are consistent with the model that the circadian clock programs the transcriptome while homeostatic process function post-trranscriptionally (*Brüning et al., 2019*; *Noya et al., 2019*), paralleling what we have found for R5. It will be of great interest to understand the circuit and molecular mechanisms by which circadian clocks regulate the R5 neuronal

calcium and synaptic properties and whether similar circuit architectures underlie daily mammalian sleep-wake.

## Materials and methods

### Fly husbandry and strains

Flies were maintained on a media of sucrose, yeast, molasses, and agar under 12:12 LD cycles at 25 °C. One- to 3-day-old female flies were separated and maintained on standard cornmeal-yeast medium under 12:12 LD cycles at 25 °C for 4 nights before experiments began. *Clk*[out] (56754), *per*ˢ (80919), *pdf-Gal4* (6899), pBDP (*pBDP-Gal4Uw*)(68384), pBDP split (*p65-AD Uw; Gal4-DBD Uw*) (79603), *R23E10-Gal4* (49032), *R69F08-Gal4* (39499), *R58H05 p59AD* (70750), *R48H04 DBD* (69353) *pdf-Gal80* (80940), *R51H05 p65AD* (70720), *R18H11 DBD* (69017), *R92H07-Gal4* (40633), *R52B02-Gal4* (38814), *R20D01-Gal4* (48889), BRPstar (55751)**,** *UAS-GCaMP6s* (42746), *UAS-TNT* (28838), *UAS-kir2.1* (6596) and *UAS-hid* (65403) were obtained from the Bloomington Drosophila Stock Center. *mGluR*-RNAi (1793) was obtained from Vienna *Drosophila* Resource Center. *MB122B* and *20xUas-IVS-Syn-GFP* was obtained from Janelia Farm.

### Behavioral assays

Following aging and entrainment, 4- to 7-day-old flies were placed in individual 5×65 mm glass capillary tubes containing sucrose-agar food (5% sucrose and 2% agar). These were then loaded into the *Drosophila* activity monitor (DAM) system (Trikinetics, Waltham, Massachusetts, USA) and placed in either an empty incubator or, in the case of SD experiments, on a multi-tube vortexer (VWR-2500) fitted with a mounting plate (Trikinetics, Waltham, Massachusetts, USA).

For SD experiments 3 nights (with 2 full days) of undisturbed sleep in 12:12 LD cycling at 25 °C served as an acclimation period and baseline. Following the baseline period, SD mechanical stimuli was performed as previously described (*Nall and Sehgal, 2013*). A 2 s vibration stimulus was applied approximately every 20 s with a randomized protocol for a time period of 2.5 hr. In the case of the forced desynchrony protocol this 2.5 hr stimulus was repeated every 7 hr (allowing for a total of 4.5 hr of rest following each stimulus) 24 times until SD occurred at each hour around the clock (*Figure 1a*). In abridged experiments, this 2.5 hr stimulus was applied 5 times: ZT0, ZT8 and ZT23 of day 3, ZT7 of day 4 and ZT6 of day 5. All behavioral experiments consist of pooled data from multiple runs with independent samples.

For sleep analyses DAM data was processed using custom MATLAB based software Sleep MAT (*Sisobhan et al., 2022*). Activity was measured in 1 min bins and sleep was identified as 5 min of inactivity (*Hendricks et al., 2000*). For SD experiments only flies deprived of >90% of baseline sleep at each SD interval were analyzed (*Pfeiffenberger and Allada, 2012*). Sleep gain was calculated as the difference between sleep during rebound and sleep during the equivalent 4.5 hr at baseline. Activity actograms were plotted with Counting Macro as previously described (*Pfeiffenberger et al., 2010a*; *Pfeiffenberger et al., 2010b*).

### Immunostaining

Following aging and entrainment, 4- to 7-day-old flies were placed in individual tubes containing sucrose-agar food (5% sucrose and 2% agar) for 3 nights. Brains were dissected in PBS (137 mM NaCl, 2.7 mM KCl, 10 mM Na2HPO4 and 1.8 mM KH2PO4) and fixed in 3.7% formalin solution (diluted from 37% formalin solution, Sigma-Aldrich) for 30 min at 4 °C. Brains were washed with 0.3% PBSTx (PBS with 0.3% Triton-X) 5 times (with 15 min shaking steps at 4 °C) before primary antibody incubation. Primary antibodies were diluted in 0.3% PBSTx with 5% normal goat serum and incubation was done at 4 °C overnight. Brains were washed for 5 times with 0.3% PBSTx. Secondary antibodies were diluted in 0.3% PBSTx with 5% normal goat serum and brains were incubated at 4 °C overnight. Primary antibody used was mouse anti-V5 (1:800 Invitrogen), Secondary antibody used was Alexa 594 anti-mouse (1:800, Invitrogen).

Images were taken using Nikon C2 confocal at ×63 magnification and acquired at 1,024 × 1,024 pixels. Analysis of BRP intensity was performed using Fiji/Imagej similarly to previously reported methods (*Liu et al., 2016*). First max intensity projections were created from confocal stacks of R5 ring projections. The mean intensity of the R5 ring was analyzed by subtracting the average intensity

of an adjacent region (background) from the average intensity of the R5 projections. Imaging data presented are derived from a single experiment due to inability to pool data from multiple experiments because of changes in laser condition and staining. All experiments were replicated a minimum of three times to confirm results.

## Intracellular Ca2$^+$ measurements

Following aging and entrainment, 4- to 7-day-old *R69F08-Gal4>UAS-GCaMP6s*, UAS-CD4-tdTomato flies were placed in individual tubes containing sucrose-agar food (5% sucrose and 2% agar) for 3 nights. Flies were dissected day 4 and imaged in ice-cold control *Drosophila* physiological saline solution (in mM: 101 NaCl, 1 CaCl$_2$, 4 MgCl$_2$, 3 KCl, 5 glucose, 1.25 NaH$_2$PO$_4$, and 20.7 NaHCO$_3$, pH 7.2, 250 mOsm) (*Flourakis et al., 2015*). Brains were held ventral side down by a harp slice grid with silica fibers from ALA scientific. GCaMP and TdTomato signal in the R5 ring neuropil was measured immediately (within 5 min) after dissection at ZT1.5 and ZT9.5. Imaging experiments were performed on an Ultima two-photon laser scanning microscope (Bruker, former Prairie Technologies, Middleton, WI). Images were acquired with an upright Zeiss Axiovert microscope with a 40×0.9 numerical aperture water immersion objective at 512 pixels ×512 pixels resolution. Single optical R5 section was selected and recorded as previously described (*Liu et al., 2016*). In brief a single optical section was selected based on visual assessment of maximum area of tdtomato signal. The GCaMP signal was recorded at ~1 fps for 60 s. The average projection of the frames was used to calculate the GCaMP and TdTomato signal.

## Connectome analysis

We accessed the NeuPrint API via R using a Natverse-based software package, *neuprintr*, along with two other open-source data visualization tools, *hemibrainr* and *ggplot2* (*Bates et al., 2020a*; *Bates et al., 2020b*). R scripts provided by the Natverse creators were modified to generate connectivity graphs (node networks) and neuron skeletonizations (visualizations of neuronal morphology). Our modified scripts can be found at https://rpubs.com/eogunlana0827/modified-code-for-analysis. Most of the neurons used in this study were identified based on their annotation in Neuprint. Cry-positive LNds were identified in the total LNd based on morphology according to the images in *Schubert et al., 2018*.

To generate node networks for sleep pathways, the body IDs of the pre- and post-synaptic targets were determined by querying the neuron types and storing the retrieved data into two data frames (A and B, respectively). Once A and B were determined, the shortest paths between the two types were then calculated. The code accounts for any duplicates that may arise when running *neuprintr*'s "shortest paths" function. This information is stored in another data frame that represents each pre- and post-synaptic neuron instance in the pathway, along with their names/types and the number of synapses between each neuron. Before establishing the network environment in which the data are plotted, the newly created data frame was modified so that only the pre- and post-synaptic neuron types and synaptic weights were included, thereby removing any body ID information. We then utilized the *network* and *ggnetwork* packages (both under the *ggplot2* package framework) to create the network environment. Colors were assigned to each neuron type using a list of variables provided in the pre-made R scripts. Finally, the connectivity graphs were plotted using *ggplot2* and exported to PDFs.

The *hemibrainr* package was used to generate visualizations of neuronal morphology from the EM data underlying Neuprint (*Bates et al., 2020a*). For each neuron type in the sleep pathways, we collected the neuron mesh data from their NeuPrint body IDs using a hemibrainr function and then stored them in a variable. Then, we randomly sampled a color to assign to each neuron type using a built-in R function. The neuron mesh was then plotted in a 3D environment, and then oriented so that the anterior side of the brain was facing the viewer.

## Fluorescence activated cell sorting and RNA-seq

FACS/RNA-seq was performed as previously reported (*Xu et al., 2019*). Briefly, flies were housed in DAM system behavior boards in either control or sleep deprivation conditions. Immediately following SD, the boards were recovered from the incubators and transferred to CO$_2$ pads. Brains were dissected in ice-cold modified dissecting saline (9.9 mM HEPES-KOH buffer, 137 mM NaCl, 5.4 mM

KCl, 0.17 mM $NaH_2PO_4$, 0.22 mM $KH_2PO_4$, 3.3 mM glucose, 43.8 mM sucrose, pH 7.4) with 0.1 µM tetrodotoxin (TTX), 50 µM D(–)–2-amino-5-phosphonopentanoic acid (AP-5), and 20 µM 6,7-dinitroq uinoxaline-2,3-dione (DNQX) to block neuronal activity. Following dissection, brains were transferred to $SM^{Active}$ medium (4.18 mM $KH_2PO_4$, 1.05 mM $CaCl_2$, 0.7 mM $MgSO_4 \cdot 7H_2O$, 116 mM NaCl, 8 mM $NaHCO_3$, 2 mg/ml glucose, 2 mg/ml trehalose, 0.35 mg/ml α-ketoglutaric acid, 0.06 mg/ml fumaric acid, 0.6 mg/ml malic acid, 0.06 mg/ml succinic acid, 2 mg/ml yeast extract with 20% non-heat-inactivated FBS, 2 mg/ml insulin and 5mM pH6.8 Bis-Tris) with 0.1 µM TTX, 50 µM AP-5, and 20 µM DNQX on ice while the rest of the brains were dissected. 40–45 brains per time point were pooled as a single sample and every condition and time point was run in triplicate for a total of twelve samples. Following dissection, the brains were pelleted by centrifugation (2000 rpm, 1 min) and washed twice with 500 µL of chilled dissecting saline (containing TTX, AP-5, and DNQX). Dissecting saline was removed and the brains were incubated at room temperature in 100 µL of papain (50 unit/mL, heat activated for 10 min at 37 °C) for 30 minutes. Following digestion, the papain was inactivated with 500 µL of chilled $SM^{Active}$ medium and then washed twice with chilled medium on ice. The brains were triturated by pipetting with a flame-rounded 1000 µL pipette tip (30 times with a medium opening, 30 times with a small opening). The sample was filtered using a 100 µm nylon filter (Sefar Nitex 03-100/32) then transferred to the Northwestern FACS core on ice. GFP-positive cells were sorted on an Aria II FACS Cell Sorter into an extraction buffer from the Arcturus PicoPure Kit. We collected 300–550 cells per sample. Following sorting, the cells were lysed in extraction buffer by incubating at 42 °C for 30 min. After lysing, the cells were stored in a –80 °C freezer until libraries could be made. 3 biological replicates for each treatment are included.

Total RNA was extracted from collected cells using the PicoPure Kit with on-column DNAse I digestion according to manufacturer instructions. Following extraction, the RNA was immediately concentrated down to 1 µL using a Speed-Vac. First strand cDNA was prepared using a T7-oligo-dT primer and SuperScript III following manufacturer instructions. Second strand synthesis was performed with DNA Polymerase (18010025), Second Strand Buffer (Cat#10812014), 10 mM dNTP (18427088), DNA Ligase (18052019), and RNaseH (18021071). The cDNA was used as a template for one round of in vitro transcription (IVT) using T7 RNA polymerase and the Ambion MegaScript kit according to manufacturer instructions. IVT was carried out at 37.5 °C for 4 hr. Following IVT, the new RNA was purified using a Qiagen RNEasy kit and then used to generate libraries for RNA-seq using an Illumina TruSeq Stranded Kit. Libraries were checked for appropriate size distribution and purity by Bioanalyzer, then sent to Novogene for sequencing. We generated 30 million reads per sample.

Reads were pseudo aligned and quantified using Kallisto (v0.46.1) (*Bray et al., 2016*) against a prebuilt index file constructed from Ensembl reference transcriptomes (v96). Kallisto was used to process paired end reads with 10 bootstraps. Differential expression analysis of the resulting abundance estimate data was then performed with Sleuth (v0.30.0; *Pimentel et al., 2017*). Gene-level abundance estimates were computed by summing transcripts per million (TPM) estimates for transcripts for each gene. To measure the effect of a particular condition against another condition for a variable, sleuth uses a Wald test which generates *p* values as well as *q* values (an adjusted p value using the Benjamini-Hochberg procedure).

## Statistics

Statistical analyses and figures were produced with Excel, Matlab and Prism. Statistical tests used, exact values of N, definitions of center, methods of multiple test correction, dispersion and precision measures and p-values are included in figure legends. Paired student T-tests were used to compare 2 groups/time points. Repeated one and two factor ANOVA analyses were used to compare multiple time points/groups with Tukey's post hoc test. Additional details regarding tests and significance values are provided in the figure legends.

## Acknowledgements

We thank the Bloomington Stock Center and the Vienna *Drosophila* Resource Center for reagents. We thank the Flow Cytometry Core and NU seq at Northwestern University for their assistance in cell sorting and sequencing. We are grateful to the members of our neighbors in the Gallio, Bass and Turek labs for their advice. This work was supported by National Institutes of Health (NIH) grant

(R01NS106955), Dept. of Army grant (W911NF1610584), Training Grants in Circadian and Sleep Research (HL790919 and HL007909), and postdoctoral NRSA grant (NS110183).

## Additional information

### Funding

| Funder | Grant reference number | Author |
|---|---|---|
| National Institutes of Health | R01NS106955 | Ravi Allada |
| Army Research Office | W911NF1610584 | Ravi Allada |
| National Science Foundation | DMS-1764421 | William Kath Ravi Allada |
| Simons Foundation | 597491-RWC | William Kath Ravi Allada |
| Northwestern University | | Clark Rosensweig |
| National Institute of Neurological Disorders and Stroke | F32 NS110183 | Clark Rosensweig |

The funders had no role in study design, data collection and interpretation, or the decision to submit the work for publication.

### Author contributions
Tomas Andreani, Conceptualization, Data curation, Formal analysis, Funding acquisition, Investigation, Methodology, Visualization, Writing - original draft; Clark Rosensweig, Conceptualization, Data curation, Formal analysis, Funding acquisition, Investigation, Methodology, Project administration, Supervision, Visualization, Writing - original draft; Shiju Sisobhan, Formal analysis, Software; Emmanuel Ogunlana, Data curation, Software; William Kath, Validation; Ravi Allada, Conceptualization, Funding acquisition, Methodology, Project administration, Supervision, Writing - original draft

### Author ORCIDs
Tomas Andreani ⓘ http://orcid.org/0000-0003-2967-9689
Clark Rosensweig ⓘ http://orcid.org/0000-0001-6364-2025
Shiju Sisobhan ⓘ http://orcid.org/0000-0002-9715-1029
Emmanuel Ogunlana ⓘ http://orcid.org/0000-0002-7955-4115
Ravi Allada ⓘ http://orcid.org/0000-0003-4371-1577

### Decision letter and Author response
Decision letter https://doi.org/10.7554/eLife.74327.sa1
Author response https://doi.org/10.7554/eLife.74327.sa2

## Additional files

### Supplementary files
• MDAR checklist

### Data availability
Sequencing data have been deposited in GEO under accession code GSE186076.

The following dataset was generated:

| Author(s) | Year | Dataset title | Dataset URL | Database and Identifier |
|---|---|---|---|---|
| Andreani T, Rosensweig C, Sisobhan S, Ogunlana E, Kath W, Allada R | 2021 | Next generation sequencing of isolated R5 ellipsoid body neurons of Drosophila in the morning and evening with and without sleep deprivation | https://www.ncbi.nlm.nih.gov/geo/query/acc.cgi?acc=GSE186076 | NCBI Gene Expression Omnibus, GSE186076 |

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
