## [Editor Report]

This work provides mechanistic insight into the interaction between the circadian and homeostatic systems that regulate sleep. Prolonged wakefulness can increase the need to sleep regardless of the time of day, so the question is whether the circadian clock influences this homeostatic regulation. The authors show that this build-up of sleep need does indeed vary with time of day under the control of specific clock neurons.

---

## [Decision Letter]

**Decision letter after peer review:**

Thank you for submitting your article "Circadian programming of the ellipsoid body sleep homeostat in *Drosophila*" for consideration by *eLife*. Your article has been reviewed by 2 peer reviewers, and the evaluation has been overseen by a Reviewing Editor and K VijayRaghavan as the Senior Editor. The following individual involved in review of your submission has agreed to reveal their identity: James Jepson (Reviewer #1).

The reviewers found your manuscript of interest, and noted in particular the innovative approach to examine circadian control of sleep rebound, the consideration of neural circuits that may contribute to time-of-day differences and the generation of gene expression data from sleep-relevant EB R5-neurons. However, there were a number of concerns about specific experiments, which could call the conclusions into question. These concerns are listed below, along with recommendations for revision:

Essential revisions:

1. Time points examined (ZTs) vary from experiment to experiment and sometimes even within an experiment e.g. Figure 3C-D (control, ZT9.5, DN1p > HID, ZT10.5; also Figure 4B-C). In general, evening points vary from 8.5 to 10.5. These should be standardized across samples, unless there is a specific reason for not doing so.

2. There is no direct evidence that the changes observed in calcium levels or gene expression in R5 Ellipsoid Body neurons are of a circadian nature, and controlled by M and E neurons. Does disruption of M and E neurons affect these molecular and physiological rhythms? Are these rhythms maintained in constant darkness, or eliminated by mutations in clock genes? While it is not necessary to show circadian regulation of all parameters measured in the EB, circadian control should be examined for at least one.

3. When discussing for example figure 6e-f, it is said that both morning and evening sleep rebound is reduced, but it is not obvious that this is the case for the evening. Can the control and experimental flies be compared directly? This is actually a question that more broadly impacts the manuscript, as several claims are made that morning and/or evening sleep homeostasis differ (or not) from control, but there are no statistics to support those claims. Statistical tests are only run within genotypes to compare time points. To make additional claims, the appropriate comparisons need to be done.

4. When manipulating neurons, only two time points are shown. Is it possible that the phase of rhythmic sleep homeostasis is altered rather than eliminated, and a rhythm is thus missed. As above, for at least one neural manipulation for each circuit tested, the whole cycle should be examined as in Figure 2.

5. The authors generally use a pBDP > transgene-X as a control for their sleep experiments. Given how easily sleep is modulated, typically Gal4 expression alone is included. While it may be unreasonable to expect that all experiments be redone with a Gal4 control, it should be added for 1-2 important circuits.

6. The authors propose that morning enhancement of sleep rebound is mediated by a DN1p > TuBu > R4m/R2 neural circuit. In this respect, it is important to show that the Glu+ DN1p neurons labelled by the R18H11/R51H05 split-Gal4 line do innervate the AOTU (where the postsynaptic sites of TuBu neurons reside). The DN1p neurons presented in Figure 3A-B look more structurally similar to the ventrally projecting DN1p subpopulation – if so, this would yield a distinct circuit hypothesis. The authors can easily check this by co-staining with BRP and using higher resolution imaging to confirm the presence of DN1p axonal tracts or presynaptic sites in the intermediate lateral region of the AOTU (e.g Lamaze et al., (2018) Current Biology, Figure 2B, D).

7. Is it possible that differences in sleep architecture between morning and evening rebound period contribute to the observed differences in rebound? For instance, the evening consists initially of high anticipation, when there is little sleep to be lost during SD, and then high sleep during the first few hours of the night, which creates a ceiling for rebound; on the other hand, there is plenty of room for sleep increase in the morning. Could this create an inherent difference between morning and evening rebound levels?

The authors are asked to at least discuss this issue.

[Editors' note: further revisions were suggested prior to acceptance, as described below.]

Thank you for resubmitting your work entitled "Circadian programming of the ellipsoid body sleep homeostat in *Drosophila*" for further consideration by *eLife*. Your revised article has been evaluated by K VijayRaghavan (Senior Editor) and a Reviewing Editor.

The manuscript has been improved but there are some remaining issues that need to be addressed, as highlighted by the reviewers below:

1. The authors were asked to confirm that projections from Glu+ DN1p neurons innervate the anterior optic tubercle (AOTU), consistent with their model that a DN1p>AOTU>R5 circuit promotes rebound in the morning.

In response, the authors provided confocal images in the new Figure 4A-C. However, the images show a posterior view of the *Drosophila* brain, and the region in which presynaptic puncta (and dendrites) of Glu+DN1p neurons are stated to innervate the AOTU (lines 227-228) does not correspond to the AOTU, which is located on the anterior of the brain adjacent to the ventral-most domain of the mushroom body α lobes (please see Figure 1A from Timaeus et al., (2020) iScience, or Figure 3A from Omoto et al., (2017) Current Biology).

The authors image is consistent with the work of Lamaze et al., (2018) Current Biology (Figure 2A), who found presynaptic termini and dendrites of R18H11-neurons in lateral posterior regions of the fly brain. But I note that these authors also showed that R18H11-DN1p dendrites do not innervate the AOTU (Figure 2C, their work).

2. Reviewers were in agreement that ablation of Glu+DN1p neurons does not cleanly show that these neurons account for differences in morning and evening rebound.

For one, although not significant, the ablation flies showed a trend towards morning gain and evening decrease. Related to this, a concern is that the lack of significant difference (P ~ 0.09) in day/night rebound shown in Figure 4J arises because the study is underpowered.

Given these concerns, we recommend that: (1) either you check the AOTU for innervation by taking anterior-focused confocal images that focus on the AOTU or revise your circuit model so it does not include the AOTU. (2) Tone down your conclusion about the role of Glu+DN1p neurons in morning rebound.

---

## [Author Response]

Essential revisions:1. Time points examined (ZTs) vary from experiment to experiment and sometimes even within an experiment e.g. Figure 3C-D (control, ZT9.5, DN1p > HID, ZT10.5; also Figure 4B-C). In general, evening points vary from 8.5 to 10.5. These should be standardized across samples, unless there is a specific reason for not doing so.

There is a specific reason for doing this noted in the second paragraph of the Results section with the following statement “To statistically compare morning and evening times of day here and throughout this study, we selected specific time points where the amount of sleep deprived and the baseline sleep during the rebound, two potential confounds, were comparable, allowing a direct comparison of sleep rebound.” Matching these variables required slightly adjusting the time points in some cases.

To address this comment, we are providing supplementary figures (for Figure 4-7) with additional time points that enable comparisons of the same time points (ZT1.5 and ZT9.5). In some cases, baseline sleep amounts now differ between ZT1.5 and ZT9.5 but sleep gain effects previously observed persist further validating our conclusions.

2. There is no direct evidence that the changes observed in calcium levels or gene expression in R5 Ellipsoid Body neurons are of a circadian nature, and controlled by M and E neurons. Does disruption of M and E neurons affect these molecular and physiological rhythms? Are these rhythms maintained in constant darkness, or eliminated by mutations in clock genes? While it is not necessary to show circadian regulation of all parameters measured in the EB, circadian control should be examined for at least one.

To demonstrate that time dependent physiological effects were due to the molecular clock, we tested if rhythmic levels of BRP were observed in the *Clk^out^* mutant background using BRPSTaR (*BRPSTaR ; Clk^out^* > *R69F08;Clk^out^* ) at ZT1.5 and ZT 9.5 (Figure 9). The time dependent difference observed in the WT background is not observed in the *Clk* mutant, thus we conclude that the observed time-dependent changes in the R5 neurons are dependent on the circadian clock.

3. When discussing for example figure 6e-f, it is said that both morning and evening sleep rebound is reduced, but it is not obvious that this is the case for the evening. Can the control and experimental flies be compared directly? This is actually a question that more broadly impacts the manuscript, as several claims are made that morning and/or evening sleep homeostasis differ (or not) from control, but there are no statistics to support those claims. Statistical tests are only run within genotypes to compare time points. To make additional claims, the appropriate comparisons need to be done.

In order to address this point, we included panels for each claim that morning or evening homeostasis are changed compared to the controls and provide corresponding statistical analyses to support these claims. These panels are available in main figures 4 and 6 and the supplemental figures of figures 3 and 7.

4. When manipulating neurons, only two time points are shown. Is it possible that the phase of rhythmic sleep homeostasis is altered rather than eliminated, and a rhythm is thus missed. As above, for at least one neural manipulation for each circuit tested, the whole cycle should be examined as in Figure 2.

In order to address this, we include the whole cycle of scheduled sleep deprivation (SSD) in the supplements of figures 3,4 and 6 for broad clock neuron drivers and for the morning (Glu^+^ DN1ps) and evening (PDF^-^ 5^TH^ sLNv and LNds) circadian circuits respectively. In each case, we do not see significant evidence for a phase shift that mask an underlying rhythm by assaying only two time points in these strains.

5. The authors generally use a pBDP > transgene-X as a control for their sleep experiments. Given how easily sleep is modulated, typically Gal4 expression alone is included. While it may be unreasonable to expect that all experiments be redone with a Gal4 control, it should be added for 1-2 important circuits.

In order to address this comment, we included the Gal4 controls for the Glu^+^ DN1ps and PDF^-^ 5^TH^ sLNv and LNs circuits in figures 4 and 6. These controls exhibit a significant difference in sleep gain between matched morning/ evening time points supporting that time dependent rebound observed in the WT flies and other controls provided are not due to genetic background effects. While we still observe time-dependent changes in sleep rebound in the controls, for one of the controls for the Glu+ DN1ps, we did not find genotype-dependent differences. We note this in the text and soften our conclusions for this line and add a caveat regarding genotype-dependent differences for other strains. However, the LNd results are validated with respect to genotype-dependent differences and this issue does not impact conclusions regarding time-dependent differences within genotype.

6. The authors propose that morning enhancement of sleep rebound is mediated by a DN1p > TuBu > R4m/R2 neural circuit. In this respect, it is important to show that the Glu+ DN1p neurons labelled by the R18H11/R51H05 split-Gal4 line do innervate the AOTU (where the postsynaptic sites of TuBu neurons reside). The DN1p neurons presented in Figure 3A-B look more structurally similar to the ventrally projecting DN1p subpopulation – if so, this would yield a distinct circuit hypothesis. The authors can easily check this by co-staining with BRP and using higher resolution imaging to confirm the presence of DN1p axonal tracts or presynaptic sites in the intermediate lateral region of the AOTU (e.g Lamaze et al., (2018) Current Biology, Figure 2B, D).

To confirm that the DN1p population we selected is in fact an anterior projecting sub-population we labeled pre and postsynaptic regions of our DN1p split line (R51H05 AD ;R18H11DBD> sytGFP; DenMark) and co-stained for BRP (Updated Figure 4a,b). Our results indicate similar projection patterns to those observed in Lamaze et al. 2018, which is expected as that paper used R18H11-Gal4 and we used R18H11-DBD and thus we are likely labeling similar cell groups. Similar to the Lamaze 2018 paper we observed presynaptic puncta both in the pars intercerebralis (PI) and more modest numbers in the AOTU. We also note significant dendritic projections to the AOTU suggesting that DN1p neurons may receive input from the TuBu neurons also.

7. Is it possible that differences in sleep architecture between morning and evening rebound period contribute to the observed differences in rebound? For instance, the evening consists initially of high anticipation, when there is little sleep to be lost during SD, and then high sleep during the first few hours of the night, which creates a ceiling for rebound; on the other hand, there is plenty of room for sleep increase in the morning. Could this create an inherent difference between morning and evening rebound levels?The authors are asked to at least discuss this issue.

We addressed this by matching morning and evening time points in which baseline sleep (both during the deprivation and during the rebound) are not significantly different. For the evening time points, the sleep rebound period includes periods of high wake when sleep rebound can be observed. Indeed, when examined graphically using our 24h SSD protocol (Figures2,3,4 and 6), one can see that rebound is observed initially but that this is quickly reversed.

[Editors' note: further revisions were suggested prior to acceptance, as described below.]

The manuscript has been improved but there are some remaining issues that need to be addressed, as highlighted by the reviewers below:1. The authors were asked to confirm that projections from Glu+ DN1p neurons innervate the anterior optic tubercle (AOTU), consistent with their model that a DN1p>AOTU>R5 circuit promotes rebound in the morning.In response, the authors provided confocal images in the new Figure 4A-C. However, the images show a posterior view of the *Drosophila* brain, and the region in which presynaptic puncta (and dendrites) of Glu+DN1p neurons are stated to innervate the AOTU (lines 227-228) does not correspond to the AOTU, which is located on the anterior of the brain adjacent to the ventral-most domain of the mushroom body α lobes (please see Figure 1A from Timaeus et al., (2020) iScience, or Figure 3A from Omoto et al., (2017) Current Biology).The authors image is consistent with the work of Lamaze et al., (2018) Current Biology (Figure 2A), who found presynaptic termini and dendrites of R18H11-neurons in lateral posterior regions of the fly brain. But I note that these authors also showed that R18H11-DN1p dendrites do not innervate the AOTU (Figure 2C, their work).

We thank the reviewers for helping us clarify the issue. We have corrected the text accordingly including removing the DN1p-Tubu connection from our figure 5 and softened the language indicating such a connection.

2. Reviewers were in agreement that ablation of Glu+DN1p neurons does not cleanly show that these neurons account for differences in morning and evening rebound.For one, although not significant, the ablation flies showed a trend towards morning gain and evening decrease. Related to this, a concern is that the lack of significant difference (P ~ 0.09) in day/night rebound shown in Figure 4J arises because the study is underpowered.Given these concerns, we recommend that: (1) either you check the AOTU for innervation by taking anterior-focused confocal images that focus on the AOTU or revise your circuit model so it does not include the AOTU. (2) Tone down your conclusion about the role of Glu+DN1p neurons in morning rebound.

We have softened our language regarding the role of the Glu+ DN1p in morning rebound as suggested.